



# Combining low and high frequency microwave radiometer measurements from the MOSAiC expedition for enhanced water vapour products

Andreas Walbröl[1], Hannes J. Griesche[2], Mario Mech[1], Susanne Crewell[1], and Kerstin Ebell[1]

[1]Institute for Geophysics and Meteorology, University of Cologne, 50969 Cologne, Germany
[2]Leibniz Institute for Tropospheric Research, Remote Sensing of Atmospheric Processes, 04318 Leipzig, Germany

**Correspondence:** Andreas Walbröl (a.walbroel@uni-koeln.de)

**Abstract.** In the central Arctic, high quality water vapour observations are sparse due to the low density of meteorological stations and uncertainties in satellite remote sensing. Different reanalyses also disagree on the amount of water vapour in the central Arctic. The Multidisciplinary drifting Observatory for the Study of the Arctic Climate (MOSAiC) expedition provides comprehensive observations that are suitable for evaluating satellite products and reanalyses. Radiosonde observations provide

high quality water vapour estimates with a high vertical but a low temporal resolution. Observations from the microwave radiometers (MWRs) onboard the research vessel *Polarstern* complement these observations through high temporal resolution. In this study, we demonstrate the high accuracy of the combination of the two MWRs HATPRO (Humidity and Temperature Profiler) and MiRAC-P (Microwave Radiometer for Arctic Clouds - Passive). For this purpose, we developed new retrievals of integrated water vapour (IWV) and profiles of specific humidity and temperature using a Neural Network approach, including

observations from both HATPRO and MiRAC-P to utilize their different water vapour sensitivity. The retrievals were trained with ERA5 data from the European Centre for Medium-Range Weather Forecasts (ECMWF) and synthetic MWR observations simulated with the Passive and Active Microwave radiative TRAnsfer tool (PAMTRA). We applied the retrievals on the synthetic and real observations and evaluated them with ERA5 and radiosondes launched during MOSAiC, respectively. To assess the benefit of the combination of HATPRO and MiRAC-P compared to single MWR retrievals, we compared the errors

with respect to MOSAiC radiosondes and computed the vertical information content of the specific humidity profiles. The root mean squared error (RMSE) of IWV was reduced by up to 15 %. Specific humidity biases and RMSE were reduced by up to 75 and 50 %, respectively. The vertical information content of specific humidity could be increased from 1.7 to 2.4 degrees of freedom. We also computed relative humidity from the retrieved temperature and specific humidity profiles and found that RMSE was reduced from 45 to 15 %. Finally, we show a case study demonstrating the enhanced humidity profiling capabilities

compared to the standard HATPRO based retrievals. The vertical resolution of the retrieved specific humidity profiles is still low compared to radiosondes but the case study revealed the potential to resolve major humidity inversions. To which degree the MWR combination detects humidity inversions, also compared to satellites and reanalyses, will be part of future work.



## 1 Introduction

The amplified warming of the Arctic, known as Arctic amplification, is a well established phenomenon and has been discussed
in several studies (e.g., Screen et al., 2012; Screen and Simmonds, 2010; Rantanen et al., 2022; Wendisch et al., 2023). Arctic
amplification is caused by several positive climate feedback mechanisms, such as the ice albedo and the lapse rate feedback
(Serreze and Barry, 2011; Wendisch et al., 2023). Following the Clausius Clapeyron relation, a warmer atmosphere can contain
more water vapour before condensation occurs. Higher water vapour loads enhance the greenhouse effect (stronger emission in
the thermal infrared) and thus increase temperatures at the surface (Held and Soden, 2000; Graversen and Wang, 2009; Ghatak
and Miller, 2013). This positive feedback loop is known as the water vapour feedback and its role in Arctic amplification is
still under investigation.

In the past decades, a moistening trend has been observed on a global scale (Chen and Liu, 2016; Allan et al., 2022) and also
regionally in the Arctic (Ghatak and Miller, 2013; Maturilli and Kayser, 2017; Parracho et al., 2018; Rinke et al., 2019; Serreze
et al., 2012). The relative increase of the vertically integrated water vapour (IWV) is strongest in the Arctic (Chen and Liu,
2016). However, IWV trends have a high spatial heterogeneity and depend on the season (Parracho et al., 2018; Rinke et al.,
2019). Many studies relied on atmospheric reanalyses, which assimilate measurements from synoptic stations, particularly
radiosondes, satellites, etc. However, ground-based observations are sparse and satellite observations have different challenges
in the Arctic (Crewell et al., 2021): The derivation of water vapour products from visible and infrared observations is hindered
by darkness or clouds, and satellite products from microwave observations are uncertain due to the high and variable sea ice
emissivity (Mathew et al., 2008; Wang et al., 2017; Scarlat et al., 2017). The lack of ground-based observations and difficulties
in satellite remote sensing in the Arctic lead to high uncertainties in water vapour products in reanalyses (Crewell et al., 2021;
Parracho et al., 2018; Chen and Liu, 2016; Graham et al., 2019b). Therefore, it is not surprising to find a large spread of the
IWV trend among reanalyses, often larger than the median trend itself for certain seasons and regions (Rinke et al., 2019).

A special feature of the Arctic is the high occurrence of humidity inversions, which are height layers where the water vapour
concentration increases with height (Devasthale et al., 2011; Vihma et al., 2011; Nygård et al., 2014; Maturilli and Kayser,
2017; Naakka et al., 2018). Humidity inversions are strongly coupled with temperature inversions (Tjernström et al., 2004),
which form due to radiative cooling in clear sky conditions in winter, or due to sea ice melt or advection of warm and moist
air above the boundary layer in summer (Graversen et al., 2008; Devasthale et al., 2010; Tjernström et al., 2019). Humidity
inversions are a moisture source for the formation and maintenance of clouds through entrainment at the cloud top (Nygård
et al., 2014). Additionally, the vertical water vapour distribution affects the downward thermal infrared radiation. Tjernström
et al. (2019) showed that in cases when humidity inversions were present, the downward thermal infrared radiation was higher
fostered by fog or low cloud formation.

Current reanalyses have difficulties in correctly representing the stable stratification of Arctic winter conditions (Wang et al.,
2019; Yu et al., 2021; Graham et al., 2019a). For example, the widely used ERA5 reanalysis from the European Centre for
Medium-Range Weather Forecast (Hersbach et al., 2020), which is among the best performing global reanalyses in the Arctic,
still shows positive near-surface air temperature and humidity biases (Graham et al., 2019a; Avila-Diaz et al., 2021; Loeb et al.,





2022; Yu et al., 2021). The biases are highest in cold stable conditions found over sea ice in winter and smaller in summer or over the open Arctic Ocean (e.g., Fram Strait, Wang et al., 2019; Graham et al., 2019b). Herrmannsdörfer et al. (2023) suggested that ERA5 does not sufficiently represent sea ice thickness and snow depth. Difficulties in the representation of the
stable conditions and positive biases of temperature and humidity at the surface result in errors in the temperature and humidity profiles of ERA5 (and other reanalyses).

It follows that reanalyses and satellite products struggle with the representation of water vapour in the Arctic. To evaluate the accuracy of water vapour in current reanalyses and satellite products, we need reference measurements. However, reliable and high quality water vapour measurements in the central Arctic are currently only available through field campaigns. The
Multidisciplinary drifting Observatory for the Study of the Arctic Climate (MOSAiC, Shupe et al., 2022) expedition, where the research vessel (RV) *Polarstern* (Knust, 2017) was frozen into the ice to observe the Arctic climate for a full annual cycle, provides unique observations for this purpose. Radiosonde measurements (Maturilli et al., 2021) yield IWV and humidity profiles with a high vertical but low temporal resolution (3–6-hourly). Additionally, water vapour products have been derived from upward looking microwave radiometers (MWRs) that were mounted on the OCEANET container (Macke et al., 2010;
Engelmann et al., 2021) at the bow of RV *Polarstern*: Walbröl et al. (2022) retrieved IWV and profiles of absolute humidity and temperature from the low frequency Humidity and Temperature Profiler (HATPRO, Rose et al., 2005) and another IWV product from the high frequency Microwave Radiometer for Arctic Clouds - Passive (MiRAC-P, Mech et al., 2019a). The MWR products have a high temporal resolution (almost every second) but the humidity profile from HATPRO is coarse with less than 2 degrees of freedom (Löhnert et al., 2009).

The high frequency observations from MiRAC-P have a high sensitivity to atmospheric water vapour in dry conditions (IWV $< 10\,\mathrm{kg\,m^{-2}}$) but get saturated in humid conditions (IWV $\geq 10\,\mathrm{kg\,m^{-2}}$, Cadeddu et al., 2007, 2022; Fionda et al., 2019). In contrast, the low frequency observations from HATPRO have a high sensitivity in humid conditions but a weak signal in the dry conditions of the Arctic in winter. The complementary moisture sensitivity of HATPRO and MiRAC-P motivates the synergy of both instruments, as it has been done for IWV in e.g., Cadeddu et al. (2009).

In this study, we develop retrievals of water vapour products combining observations from HATPRO and MiRAC-P to improve the vertical resolution of specific humidity profiles and reduce errors compared to single MWR retrievals. We retrieved specific humidity instead of absolute humidity because it is a more commonly used humidity measure in atmospheric reanalyses and satellite products. Specifically, we answer the following questions:

1. How much are IWV and humidity profile errors reduced compared to single instrument retrievals and what is the influ-
ence of using different retrieval setups?

2. What is the vertical information content benefit for humidity retrievals when combining two MWRs with different moisture sensitivity?

3. Is the vertical information content sensitive to cloud presence, temperature or water vapour amount?

The manuscript is structured as follows: In Sect. 2, we start with a description of the data used for the retrieval development
and the measurements from the MOSAiC expedition, which will be used for the application and evaluation of the retrieval.



In Sect. 3, we elaborate on the preparation of the retrieval development data before giving details on the retrieval setup and vertical information content estimation. Afterwards, we evaluate the retrieval in Sect. 4 and estimate the information benefit in Sect. 5 before concluding the manuscript in Sect. 6 by answering the questions raised above.

## 2 Data sets

### 2.1 Retrieval development data

Radiosondes are commonly used for the evaluation of temperature and humidity profile retrievals because of the high vertical resolution and accuracy (e.g., Cimini et al., 2010; Löhnert and Maier, 2012). Due to the lack of radiosonde stations and uncertain water vapour observations from satellites, we selected the ERA5 reanalysis (Hersbach et al., 2020) as a data source for the retrieval development. With a horizontal resolution of 31 km and 137 vertical levels, it has the highest horizontal and vertical resolution of all current global reanalyses. The high vertical resolution might be beneficial for developing humidity profile retrievals because a low vertical resolution could constrain the retrieval from reaching its true potential. ERA5 data is available for 1940–present with an hourly resolution. Despite having slightly higher biases in near-surface air temperatures and humidity in cold stable conditions over sea ice than other reanalyses, ERA5 overall performs best in the Arctic, especially concerning the representation of clouds and precipitation (Graham et al., 2019a). The better representation of clouds and precipitation is beneficial for the simulation of microwave radiances for the retrieval development (described in Sect. 3.1). Also, extreme precipitation and temperature events are better captured by ERA5 than other reanalyses (Avila-Diaz et al., 2021; Wang et al., 2019; Loeb et al., 2022).

### 2.2 MOSAiC observations for retrieval application and evaluation

RV *Polarstern* drifted with an ice floe from 04 October 2019 in the Laptev Sea across the central Arctic Ocean until it approached the marginal ice zone in the Fram Strait on 31 July 2020. Between mid-May and mid-June 2020, RV *Polarstern* had to leave the floe for logistical reasons. To capture the refreezing period of the ice, RV *Polarstern* drifted with a second ice floe close to the North Pole from 21 August to 20 September 2020. In early October 2020, RV *Polarstern* left the sea ice.

### 2.2.1 Radiosondes

Throughout MOSAiC, Vaisala RS41 radiosondes have been launched from RV *Polarstern* at the standard synoptic times (00, 06, 12, and 18 UTC). The actual launch time is usually around 1 hour before the respective synoptic time due to the relatively slow ascent rates of about $5 \, \mathrm{m \, s^{-1}}$. During intense observation periods, additional radiosondes were launched at 03, 09, 15 and 21 UTC. Here, we use all radiosonde level 2 data from 01 October 2019 to 01 October 2020 (Maturilli et al., 2021). Radiosondes provide temperature, pressure, and relative humidity with accuracies of 0.2–0.4 K, 0.04–1.0 hPa, and 3–4 %, respectively. With a measurement frequency of 1 Hz, the vertical resolution is about 5 m. For the comparison with the retrievals, the radiosonde data has been interpolated onto the retrieval height grid (see Sect. 3.1).





### 2.2.2 Cloudnet and surface meteorology measurements

To evaluate the presented retrievals in different atmospheric conditions, we included additional data sets from the MOSAiC expedition: To distinguish between freezing and non-freezing conditions at the surface (temperatures below and above 273.15 K), the 2 m temperature measurements from the tower at the Met City site (Cox et al., 2023) were used. The Met City site was
located within the central observatory, only a few hundred metres away from RV *Polarstern*. Additionally, we identified cloudy scenes using the Cloudnet retrieval products (Griesche et al., 2024, *accepted*). Cloudnet uses a synergy of passive and active atmospheric remote sensing to provide profiles of cloud macro- and microphysical properties (liquid and ice water content, effective radii of liquid droplets and ice crystals) with a time and height resolution of 30 s and 30 m, respectively (Illingworth et al., 2007; Tukiainen et al., 2020).

Cloudnet delivers, e.g., a classification of the atmospheric conditions, distinguishing between clear sky, different cloud types (ice, liquid, mixed-phase), and the presence of aerosols and insects, for each time-height pixel. Because of technical limitations, the Cloudnet product starts at a height of 182 m and can therefore miss the presence of low-level stratus clouds, which are common in the Arctic (Gierens et al., 2020; Griesche et al., 2020). The additional low-level stratus detection developed by Griesche et al. (2020) was used to mask these cases.

In this study, clear sky conditions were identified using Cloudnet target classification data (Engelmann et al., 2023) and the low-level stratus mask (Griesche et al., 2023) where quality flags indicated good quality (including also the Cloudnet issue dataset, Griesche and Seifert, 2023). As we compare our retrievals with radiosonde measurements, we selected Cloudnet data at times from the radiosonde launch to 15 minutes after the launch. A radiosonde launch is considered clear sky when no low-level stratus were present and the Cloudnet target classification indicated either clear sky, aerosols, or insects.

### 140 2.2.3 Microwave radiometers

The two upward-looking microwave radiometers HATPRO and MiRAC-P measure radiation emitted from water vapour, oxygen and hydrometeors. Measured radiances are typically expressed as brightness temperatures (TB). HATPRO detects radiances in seven channels between 22.24 and 31.4 GHz (K–band) and in seven channels between 51.26 and 58 GHz (V–band). MiRAC-P has a double-sideband receiver that measures radiances at six frequencies from 183.31±0.6 to 183.31±7.5 GHz (G–
band) and a two-channel receiver for 243 and 340 GHz. At MiRAC-P frequencies, the scattering of radiation by hydrometeors is relevant, and the contribution of the continuum water vapour absorption is stronger (Rosenkranz, 1998).

Figure 1 shows TBs simulated with the Passive and Active Microwave radiative TRAnsfer tool (PAMTRA, Mech et al., 2020), using two clear sky radiosondes from MOSAiC (winter: 05 March 2020, 06 UTC, summer: 06 August 2020, 00 UTC). A higher atmospheric opacity generally results in higher TBs in the zenith. In the K–band channels of HATPRO and the G–
band channels of MiRAC-P, which are located around resonant water vapour absorption lines, the different water vapour loads of winter and summer can be well distinguished by their large TB differences of up to 40 K in the K–band and more than 100 K in the G–band. Also in MiRAC-P's high frequency channels at 243 and 340 GHz, TB differences are larger than at K–band



frequencies (up to 200 K) due to continuum water vapour absorption. At the K–band frequencies, the relation between TBs and IWV is rather linear and becomes more nonlinear for the higher frequencies (G–band and above).

Observations along resonant water vapour absorption lines are well suited to derive IWV and humidity profiles (Crewell et al., 2001; Cadeddu et al., 2007; Cimini et al., 2010; Perro et al., 2016, e.g.,). Because of the high water vapour sensitivity, most of the G–band channels are saturated in the summer case, meaning they do not observe radiances from the entire atmospheric column. In contrast, the K–band channels show almost no water vapour signal in the extremely dry winter case (IWV of $0.9\,\mathrm{kg\,m^{-2}}$) while there is still a strong signal in the G–band.

Furthermore, higher TBs in summer compared to winter are caused by higher temperatures of the emitting gases and hydrometeors (Fig. 1). The V–band channels of HATPRO lie around the oxygen absorption complex and can be used for temperature profiling (Rose et al., 2005; Löhnert and Maier, 2012). As explained in Walbröl et al. (2022), HATPRO also measured atmospheric radiances at different elevation angles every 30 minutes during MOSAiC, allowing for more detailed temperature profile retrievals in the lower troposphere (boundary layer temperature profiles).

In this study, we generally used TB measurements where flags indicate good quality (Walbröl et al., 2022). We identified a few rain events between late-May and late-June 2020 that were not flagged by visual inspection. The quality flags have been updated. Additionally, we checked whether other flag values could be accepted and found that a receiver sanity flag was often set although the data looked reasonable. Therefore, we also included that data in our analysis. Times before the first successful calibration of both MWRs (22 October 2019, 05:40 UTC) have been excluded.

For the information benefit analysis, we compared the new synergistic retrievals to the single instrument retrievals developed by Walbröl et al. (2022), i.e., the two IWV products (HATPRO and MiRAC-P), and profiles of temperature and absolute humidity from HATPRO (Ebell et al., 2022; Walbröl et al., 2022b). We converted the retrieved absolute to specific humidity using the retrieved temperature profiles and air pressure from radiosondes. All retrieved quantities were averaged over 15 minutes, starting at the launch time of each radiosonde, for the comparison with MOSAiC radiosondes. For boundary layer

temperature profiles, we extended the averaging window to 30 minutes before to 30 minutes after each radiosonde launch due to the lower sampling rate.

## 3   Methods

The retrieval of an atmospheric state vector $\boldsymbol{x}$ (e.g., specific humidity profile) from an observation vector $\boldsymbol{y}$ (e.g., TBs at different frequencies) is an inverse problem. In its simplest form, the inverse problem can be formulated as $\boldsymbol{x} = F^{-1}(\boldsymbol{y})$ where

$F$ is the forward operator (e.g., radiative transfer model, here, PAMTRA). In atmospheric remote sensing, inverse problems are often ill-conditioned because small changes in observations can lead to large changes in the retrieved state vector and many different atmospheric states can lead to the same observations. Furthermore, the inverse problem is ill-posed because the radiative transfer equation cannot be inverted in a direct way.

The challenge is to find the most probable and realistic state of the atmosphere that fits the observations. In physical re-

trievals (e.g., Optimal Estimation, Rodgers, 2008; Ebell et al., 2017), the state vector $\boldsymbol{x}$ is adapted as long as the forward sim-



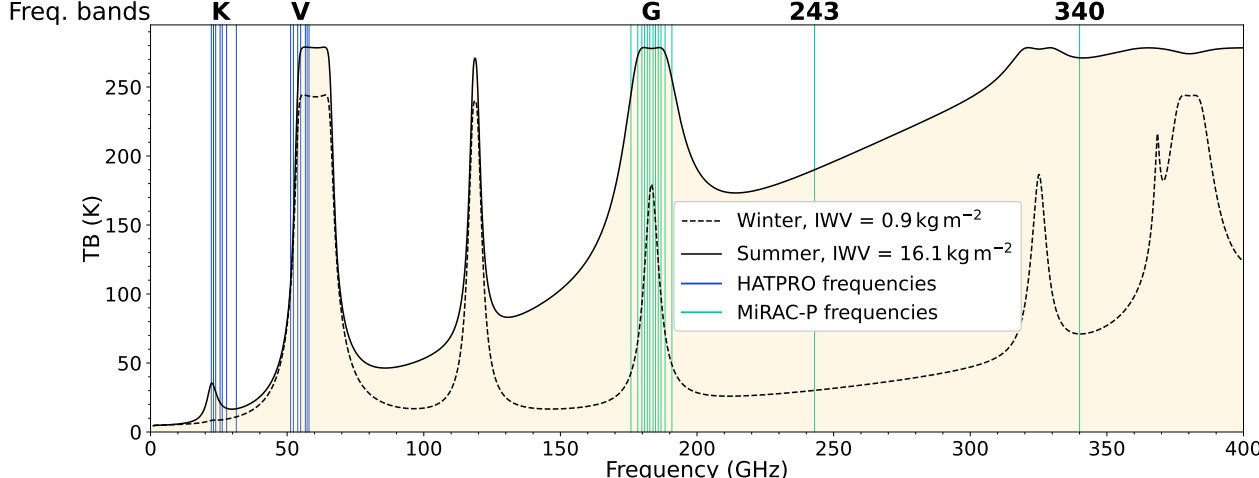

**Figure 1.** Brightness temperatures (TBs) from 1 to 400 GHz simulated with PAMTRA for two radiosondes launched from RV *Polarstern* during MOSAiC (winter: 05 March 2020, 06 UTC, summer: 06 August 2020, 00 UTC). The dashed (solid) black line shows the TBs simulated with meteorological data from the winter (summer) radiosonde. The blue (cyan) lines indicate the frequencies at which HATPRO (MiRAC-P) measures. The labels K, V, G, 243, and 340 represent abbreviations for sets of frequency channels (bands) of HATPRO and MiRAC-P.

ulated observations $F(\boldsymbol{x})$ do not agree with the actual observations $\boldsymbol{y}$ within a given uncertainty range. Physical retrievals are computationally expensive but provide physically consistent state vectors and uncertainty estimation. Computationally cheap approaches that are also well established and provide similarly good results are statistical retrievals (Solheim et al., 1998). In statistical retrievals, empirical relations are used to map observations to the state vector. The statistical relationship between observations and state vector must be trained with large data sets covering the conditions of the area of interest. Regression or deep learning algorithms are examples of statistical retrievals. In this study, we use Neural Networks (NNs) because they can deal better with the nonlinear relationship between IWV and TB measurements in the G–band.

### 3.1 Retrieval preparation

For the NN retrievals of IWV, specific humidity and temperature profiles during MOSAiC, a training data set is needed that covers the variability of the environmental conditions in the central Arctic over an annual cycle. We selected ERA5 data for 2001–2018 with 6-hourly temporal resolution at 12 grid points, of which 9 are located in the central Arctic and 3 in the Fram Strait (see Fig. 2). The grid points in the Fram Strait cover more humid conditions as this is a typical pathway for warm and moist air intrusions (Mewes and Jacobi, 2019).

Simulated HATPRO and MiRAC-P observations are needed in conjunction with the ERA5 data to train the NN. Meteorological data (temperature, relative humidity, geopotential height, pressure, 10 m wind) and vertical hydrometeor distributions from ERA5 (specific cloud liquid, ice, rain and snow content) have been used as input to simulate TBs with PAMTRA. The



ERA5 skin temperature was used for the sea ice and sea surface temperatures. The TBs were simulated with PAMTRA's default gaseous absorption, hydrometeor absorption and scattering models as described in Mech et al. (2020).

Four years of simulated TBs and ERA5 data (2001, 2006, 2011, and 2015) were held back from the retrieval development
for the final evaluation (ERA5 evaluation data set). With the remaining 14 years of data, we trained the NN and validated its performance (11 and 3 years for the training and validation data sets, respectively). The number of training (validation) samples is roughly 192000 (52000). To avoid training near-surface temperature and humidity biases from ERA5 into the retrieval, a small subset of about 5% of level 2 MOSAiC radiosondes (Maturilli et al., 2021) was also included in the validation process. For the retrieval development and evaluation, atmospheric profiles have been interpolated onto the same height grid used
in Walbröl et al. (2022), ranging from 0 to 10000 m with the vertical spacing increasing from 50 m at the surface to 500 m at the top. The height grid was limited to 8000 m for temperature profiles to avoid the tropopause. Additionally, to imitate measurement uncertainties, random Gaussian noise with a mean of 0 and standard deviations of 0.5, 0.75, and 2.5 K has been added to the simulated TBs at K–V, G, and 243–340 GHz, respectively. We intentionally used a higher noise level for the higher frequencies to account for the higher PAMTRA simulation uncertainties due to scattering from hydrometeors and water vapour
continuum absorption.

## 3.2 Retrieval setup

This study used multilayer perceptron NNs (fully connected layers) to retrieve IWV, specific humidity, and temperature profiles. To optimally use HATPRO's boundary layer observations, we retrieved temperature profiles from zenith and boundary layer observations separately. The challenge is to develop retrievals that are not overfitted and can therefore adapt well to new data.
Overfitting occurs when the retrieval does not only learn the relation between the observations and the atmospheric state but also the (synthetic) noise. Additionally, we wanted to ensure that the retrievals are robust by training an ensemble of 20 NNs with identical settings but with different random number seeds. The random number seeds affect the selection of years for the training and validation data, as well as the NN initialization (weight coefficients). The NNs are considered robust when the errors in the validation data show a small spread over the ensemble of 20 NNs. For example, the spread should be smaller than
a given threshold (e.g., $0.2 \, \mathrm{kg \, m^{-2}}$ for IWV) or smaller than the magnitude of the error.

To meet the retrieval performance requirements, we developed four NNs with different settings (see Appendix A), one for each retrieved quantity (IWV, specific humidity, and temperature profiles from zenith and boundary layer observations). The retrievals of profiles required deeper networks and stronger regularization measures (e.g., dropout layers, batch normalization, see Appendix A) to avoid overfitting and to achieve a good performance. Besides TBs at different frequencies, we also included
seasonal information in the form of the cosine and sine of the day of the year as input to all NNs except for the boundary layer temperature profile (inspired by Billault-Roux and Berne, 2021). Additionally, adding the 2 m temperature and the retrieved IWV as input to the specific humidity profile retrieval slightly reduced errors during validation. Therefore, the specific humidity retrieval can only be performed after the IWV retrieval. For the boundary layer temperature profile, the input vector consists of V–band TBs at various elevation angles (90.0, 30.0, 19.2, 14.4, 11.4, 8.4, 6.6 and 5.4°), which are measured during HATPRO's
boundary layer scan. TBs at other frequencies were not included because they were not measured at these elevation angles.





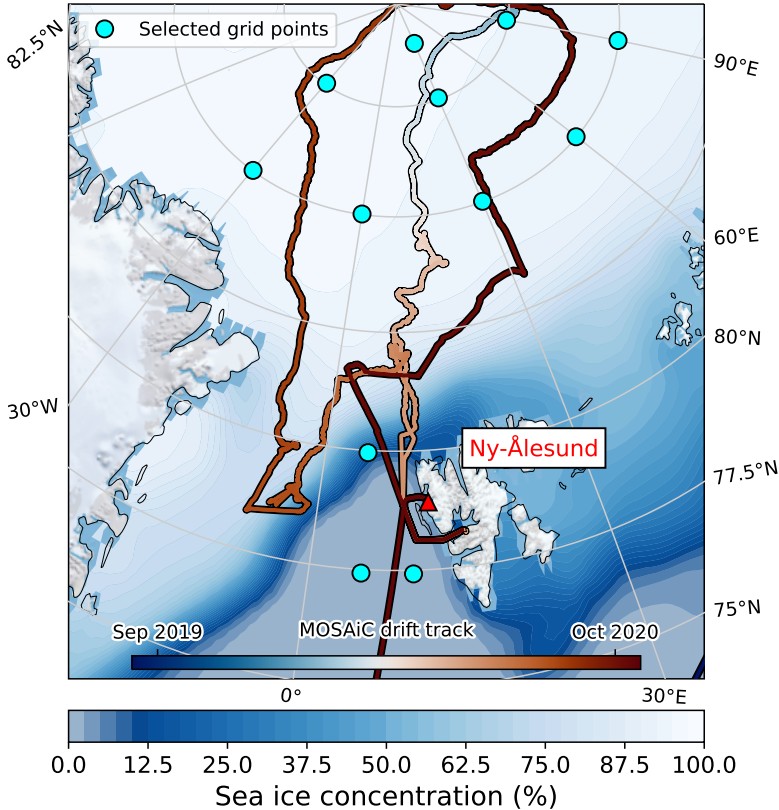

**Figure 2.** Mean sea ice concentration in the Arctic over the years 2001—2018 based on daily ERA5 data at 12 UTC. Light blue circles mark the position of the 12 grid points selected for the retrieval development. The MOSAiC drift track is marked as coloured line with black outline.

Also, adding other parameters to the input vector did not improve errors. Therefore, the input vector is identical to the one used in the HATPRO regression retrieval described in Walbröl et al. (2022). Further details of the NN retrieval principles and settings can be found in Appendix A.

### 3.3 Metrics for retrieval evaluation and vertical information content

The retrieved state vector $x$ (e.g., specific humidity profile) is evaluated using the reference $\tilde{x}$ provided by ERA5 (ERA5 evaluation data set) or MOSAiC radiosondes (MOSAiC evaluation data set). For each component $j$ of the state vector (i.e.,



$j$-th height level), we calculate the bias, the root mean squared error (RMSE) and the bias-corrected RMSE:

$$\mathrm{Bias}_j = \frac{1}{N_s} \sum_{i=0}^{N_s} (x_{ij} - \tilde{x}_{ij}) \tag{1}$$

$$\mathrm{RMSE}_j = \sqrt{\frac{1}{N_s} \sum_{i=0}^{N_s} (x_{ij} - \tilde{x}_{ij})^2} \tag{2}$$

$$\mathrm{RMSE}_{\mathrm{corr}\,j} = \sqrt{\frac{1}{N_s} \sum_{i=0}^{N_s} ((x_{ij} - \mathrm{Bias}_j) - \tilde{x}_{ij})^2} \tag{3}$$

$N_s$ is the number of data samples of the respective evaluation data set. For IWV, we also compute the Pearson product-moment correlation coefficient

$$\mathrm{R} = \frac{\sum_{i=0}^{N_s} (\tilde{x}_i - \bar{\tilde{x}})(x_i - \bar{x})}{\sqrt{\sum_{i=0}^{N_s} (\tilde{x}_i - \bar{\tilde{x}})^2 \sum_{i=0}^{N_s} (x_i - \bar{x})^2}}, \tag{4}$$

where $\bar{x}$ ($\bar{\tilde{x}}$) is the mean retrieved (reference) state vector.

The vertical information content of passive microwave observations was computed following the ideas of physical retrievals of Rodgers (2008). Due to computation time, the information content was only computed for a randomly selected subset of $4\,\%$ of the ERA5 evaluation data set (2803 samples). Firstly, we interpolated the vertical grid from the ERA5 model levels to the retrieval height grid and simulated new reference observation vectors $\boldsymbol{y}$ (here, TBs) with PAMTRA. For these simulations, the retrieval grid has been extended to $45000\,\mathrm{m}$ to simulate emissions from gases (mainly oxygen) beyond the retrieval height grid. Secondly, each state vector component is perturbed step by step. We multiply the respective height level by 1.01 for specific humidity profiles, similar to Ebell et al. (2013). Thirdly, we simulate new TBs with PAMTRA for each perturbed state vector. Fourthly, the Jacobian Matrix $\mathbf{K}$ is calculated with entries $K_{aj} = \partial y_{ia}/\partial x_{ij}$ where $\partial y_{ia}$ is the $a$-th component of the difference between the perturbation-based and reference observation vector of the $i$-th data sample. $\partial x_{ij}$ is the $j$-th component ($j$-th height level) of the difference between the perturbed and reference state vector. Fifthly, the Averaging Kernel matrix $\mathbf{A}$ is computed with $\mathbf{A} = \left(\mathbf{K}^{\mathrm{T}} \mathbf{S}_\varepsilon^{-1} \mathbf{K} + \mathbf{S}_a^{-1}\right)^{-1} \mathbf{K}^{\mathrm{T}} \mathbf{S}_\varepsilon^{-1} \mathbf{K}$ where $\mathbf{S}_a$ and $\mathbf{S}_\varepsilon$ are the covariance matrices of the state and observation vectors, respectively. $\mathbf{S}_\varepsilon$ contains the TB noise on the main diagonal while the remaining entries are 0. $\mathbf{S}_a$ is calculated as full covariance matrix from the ERA5 evaluation data set. Finally, the degrees of freedom (DOF) are inferred from the trace of the Averaging Kernel $\mathbf{A}$.

## 4 Retrieval evaluation

We applied the retrievals to both the ERA5 evaluation data set and MOSAiC observations (MOSAiC evaluation data set), for which the radiosondes serve as the reference data set. The retrieval evaluation with respect to the ERA5 data allows us to assess the retrievals' theoretical best performance because it is an idealized world without measurement problems. Here, we compute errors for all 20 NNs to get an idea of the spread among the NNs. For the evaluation with the MOSAiC radiosondes,





we selected the NN that has a low RMSE and bias in the validation data set while also having the lowest RMSE in the 5 %
MOSAiC radiosonde subset that we included in the validation process. Hereafter, this NN is referred to as the final NN.

## 4.1 IWV

The performance of the IWV retrieval applied to the ERA5 and MOSAiC evaluation data sets can be seen in Fig. 3. For the
ERA5 data, we can evaluate the robustness of the NN through the spread of the errors among all 20 NNs. The RMSE of IWV
varies little over the 20 NNs for IWV up to $24\,\mathrm{kg\,m^{-2}}$, indicated by the low spread ($< 0.3\,\mathrm{kg\,m^{-2}}$). Only for higher IWV, the
spread increases significantly to $0.8\,\mathrm{kg\,m^{-2}}$. However, only 41 of 70080 ($< 0.1\,\%$) of the synthetic data set samples have an
IWV above $24\,\mathrm{kg\,m^{-2}}$. Therefore, errors are computed over a very low fraction of the data and tend to vary more for different
NNs. Most importantly, statistical retrievals such as NNs struggle to capture extreme conditions not well represented in the
training data set. This can also be seen in the bias, which is close to zero for IWV below $20\,\mathrm{kg\,m^{-2}}$ as expected for a well
trained NN, but deviates from zero for higher IWV. However, biases are still small for both the ERA5 and MOSAiC evaluation
data sets, staying below 2 %.

The RMSE of the final NN, which was selected based on errors in the validation data set, is about 2 % of the IWV, and
therefore also at the lower end of the 20 NN ensemble for the ERA5 evaluation data set. This shows that the retrieval is well
trained because it performs similarly well on the evaluation data set as on the validation data set. For the comparison with
MOSAiC observations, where we also use the final NN, the RMSE is slightly higher in most IWV regimes, reaching up to
3–4 %. In absolute terms, the RMSE increases from 0.1 to $0.7\,\mathrm{kg\,m^{-2}}$ with IWV increasing from 1 to $29\,\mathrm{kg\,m^{-2}}$. Here, the
additional uncertainties in the radiosonde measurements and matching with the MWR data must be considered.

## 4.2 Specific humidity profiles

We evaluate the retrieved specific humidity profiles ($q$) in terms of bias and $\mathrm{RMSE_{corr}}$ for the ERA5 and MOSAiC evaluation
data sets (Fig. 4). The RMSE values are similar to $\mathrm{RMSE_{corr}}$ because of a small bias. For the MOSAiC data, the $\mathrm{RMSE_{corr}}$
increases from $0.25\,\mathrm{g\,kg^{-1}}$ at the surface to $0.5\,\mathrm{g\,kg^{-1}}$ at $1500\,\mathrm{m}$, which is 15 to 30 % of the mean specific humidity (Fig. 4b). At
higher altitudes, the $\mathrm{RMSE_{corr}}$ is lower but the relative error increases because the mean specific humidity also decreases. While
the $\mathrm{RMSE_{corr}}$ are generally smaller for the ERA5 data, the shape is similar with the highest $\mathrm{RMSE_{corr}}$ of about $0.25\,\mathrm{g\,kg^{-1}}$
(15 % of the mean $q$) at $1000\,\mathrm{m}$ and even lower values at the surface with $0.15\,\mathrm{g\,kg^{-1}}$ (8 %). The $\mathrm{RMSE_{corr}}$ spread across all 20
NNs is negligible, mostly ranging from 0.01 to $0.02\,\mathrm{g\,kg^{-1}}$.

The mean MOSAiC radiosonde $q$ profile shows the maximum value about $250\,\mathrm{m}$ lower than the mean retrieved $q$ profile
(Fig. 4). Because of the different heights of the humidity inversion, we find the highest $\mathrm{RMSE_{corr}}$ and bias slightly above
the height level of the maximum $q$ value (at $1500\,\mathrm{m}$). At this height, the retrieved $q$ profile overestimates the radiosonde
measurement by up to $0.15\,\mathrm{g\,kg^{-1}}$ (see bias in Fig. 4a). Above $3500\,\mathrm{m}$, the bias remains negative with values up to $-0.04\,\mathrm{g\,kg^{-1}}$
at $5500\,\mathrm{m}$. On the ERA5 evaluation data set, the final NN, which was also used to derive the $q$ profile for MOSAiC, denotes
much smaller biases and is slightly negative for all heights (only up to $-0.025\,\mathrm{g\,kg^{-1}}$). However, in the lowest $2000\,\mathrm{m}$, the bias
varies much more than the $\mathrm{RMSE_{corr}}$, ranging from $-0.1$ to $+0.1\,\mathrm{g\,kg^{-1}}$ depending on the chosen NN.



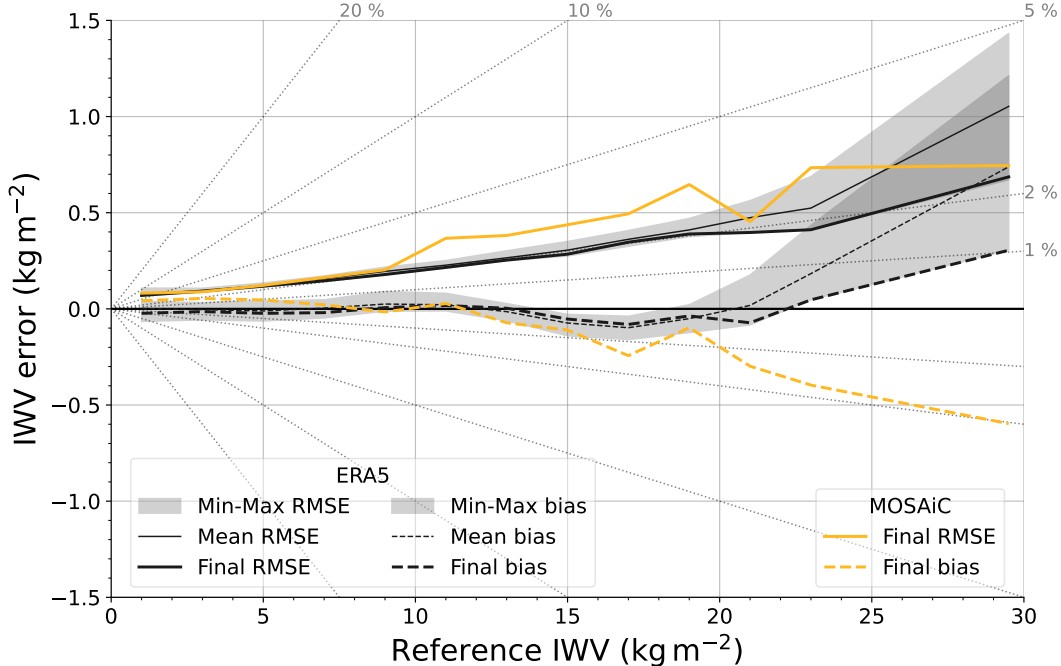

**Figure 3.** IWV errors (RMSE and bias) for certain bins of reference IWV (0–2, 2–4, ..., 22–24, 24–35 $\mathrm{kg\,m^{-2}}$). IWV errors based on the ERA5 (MOSAiC) evaluation data set are displayed in black (yellow). The maximum and minimum spread of RMSE and bias over the 20 Neural Networks are indicated by grey shading. The RMSE (bias) of the mean over the 20 Neural Networks is displayed as a thin solid (dashed) black line. The RMSE (bias) of the final NN is shown as a thick solid (dashed) black line.

The smaller magnitude of the error profiles in the ERA5 evaluation data set is likely due to the lower complexity of $q$ profiles in ERA5 compared to radiosonde observations. Specific humidity profiles in reanalyses are typically much smoother and do not resolve small inversions (Chellini and Ebell, 2022). Passive microwave observations cannot resolve small inversions and

average out strong vertical gradients. Therefore, errors of retrieved profiles are large when compared to radiosonde data in the presence of strong vertical gradients (i.e., humidity inversions), while the smoother profiles of reanalyses can be captured better. As the retrieval has been trained with reanalysis data, it is also expected to perform best when applied to the same reanalysis. Furthermore, the errors of the evaluation based on real observations can be higher due to measurement errors of radiosondes (noise, sonde drift, systematic errors due to sensor response time, etc.) and of the MWRs (noise, systematic errors).

**4.3 Temperature profiles**

For the evaluation of the retrieved temperature profiles, we also analyze the bias and $\mathrm{RMSE_{corr}}$ (Fig. 5) but distinguish between profiles retrieved from zenith observations (henceforth, zenith temperature profiles) and boundary layer scan (henceforth, BL temperature profiles). As for specific humidity, the spread over the 20 NNs is larger for the bias than for $\mathrm{RMSE_{corr}}$ but generally quite small (especially for BL temperature profiles).



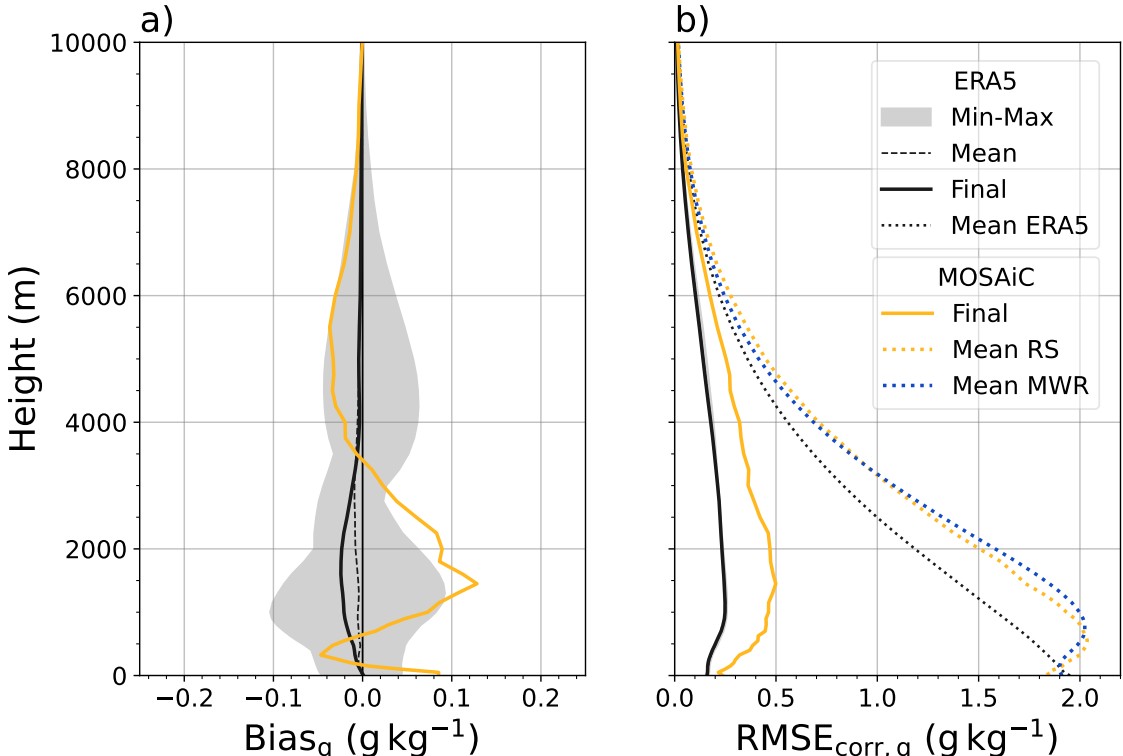

**Figure 4.** Specific humidity $q$ error profiles showing (a) the bias and (b) the bias-corrected RMSE with respect to the reference from the ERA5 and MOSAiC evaluation data sets. The dashed black line in each panel shows the mean over the 20 Neural Networks while shading indicates the min–max spread. The prediction of the final Neural Network is indicated by the thick black (yellow) lines for the ERA5 (MOSAiC) evaluation data set. The mean MOSAiC radiosonde (RS) profile and ERA5 profile are shown as yellow and black dotted lines, respectively, and serve as reference for the absolute error values. The mean retrieved profile from MOSAiC microwave radiometer observations (MWR) is also included as blue dotted line.

Firstly, we evaluate the zenith temperature profiles: The biases and $RMSE_{corr}$ of zenith temperature profiles are larger for the MOSAiC compared to the ERA5 evaluation data set below 1500 m but mostly similar at higher altitudes (see Fig. 5a and b). Within the lowest 150 m, the MOSAiC data $RMSE_{corr}$ decreases rapidly from 2.9 to 1.4 K. This large $RMSE_{corr}$ is associated with near-surface temperature inversions that typically occur in the Arctic. In the ERA5 evaluation data set, this steep error gradient is less pronounced because near-surface temperature inversions over sea ice are not well represented in ERA5. Between about 200 and 2000 m, the $RMSE_{corr}$ is between 1.2 and 1.6 K for the MOSAiC and between 0.8 and 1.6 K for the ERA5 evaluation data set. At the top of the retrieval grid at 8000 m, the $RMSE_{corr}$ increases to 2.5 K for MOSAiC and 3 K for ERA5.

In the lowest 500 m, the bias of the zenith temperature profiles lies between $-1$ and $+1$ K for the MOSAiC and between $-0.2$ and $+0.2$ K for the ERA5 evaluation data set (final NN, see Fig. 5a). Here, also the strong surface temperature inversions,





which are not well resolved by the retrieved profile, are responsible for the large bias. Above 1500 m, the bias in both data sets is generally smaller than $\pm 0.2$ K. However, the MOSAiC observation bias varies over the seasons: In winter (22 October 2019–30 April 2020), the bias is mostly negative in the mid-troposphere, ranging from $-0.4$ to $-0.8$ K, while they are positive in summer (01 May–01 October 2020), ranging from $+0.5$ to $+0.9$ K (not shown).

As expected, biases and RMSE$_{\text{corr}}$ are smaller for the BL temperature profiles in the lowest 1500 m compared to the zenith temperature profiles (see Fig. 5c and d). This result is consistent with the findings of Crewell and Löhnert (2007). For the MOSAiC data, the RMSE$_{\text{corr}}$ is 2 K at the surface (0.9 K at 100 m) and smaller than 1.2 K up to 1 km height. The error is therefore 1 K (0.5–0.6 K) lower compared to the zenith temperature profile error. Based on the ERA5 evaluation data set, the near-surface RMSE$_{\text{corr}}$ values are only 0.4–0.5 K, which is lower than for the MOSAiC data because of the less complex temperature profile and the absence of measurement uncertainties. In the lowest 1500 m, also the bias is reduced, being nearly 0 K in the ERA5 evaluation data set (with the final NN), and between $-0.6$ and $+0.4$ K in the MOSAiC data. Also, the seasonal variation of the MOSAiC BL temperature profile bias is smaller than that of the zenith temperature profiles. Above 2000 m, the RMSE$_{\text{corr}}$ is similar for both the zenith and BL temperature profiles but the bias above 2000 m is stronger (more negative) in BL temperature profiles, especially for the MOSAiC data (up to $-2$ K).

We conclude that if the 30 minute temporal resolution is sufficient for the user, a combination of BL profiles and zenith profiles provides optimal performance. We recommend that BL temperature profiles should be used in the lowest 1500 m, followed by a linear transition to the zenith temperature profile between 1500 and 2000 m and only the zenith temperature profile above 2000 m.

## 5 Information benefit analysis

After introducing the combined HATPRO and MiRAC-P retrieval, it still has to be demonstrated that the synergy is beneficial compared to single instrument retrievals. The benefit is quantified through error reduction and gain in vertical information content. We compare the errors of the synergy with the single instrument retrievals by Walbröl et al. (2022) for MOSAiC observations to present the improvements for actual observations. As the retrieval methods also differ, we also analyzed the influence of different retrieval architectures (i.e., NN instead of regression) and training data sets (ERA5 instead of Ny-Ålesund radiosondes) on the error reduction compared to HATPRO-only retrievals. This helps to isolate the pure benefit of the combination of low and high frequency microwave observations from potential effects due to different retrieval methods. In Sect. 5.1 and 5.2, the error estimates for the synergy correspond to the ones shown with respect to MOSAiC radiosondes in Sect. 4.1 and 4.2.

### 5.1 IWV

Figure 6 shows the RMSE and bias of IWV obtained from single instrument observations (HATPRO-only, MiRAC-P-only) and from the synergy of both instruments, with radiosonde IWV as reference. As found in Walbröl et al. (2022), the HATPRO-only IWV retrieval denotes high relative errors and a positive bias ($> 20\,\%$) for IWV below $5\,\text{kg m}^{-2}$, while having lower relative





**Figure 5.** Error profiles of (a,b) zenith and (b,d) boundary layer temperature $T$ profiles. Panels (a) and (c) show the bias and panels (b) and (d) the bias-corrected RMSE with respect to the reference from the ERA5 and MOSAiC evaluation data sets. Shading and different line types are similar to Fig. 4.





errors (2–4 %) for IWV greater than $10\,\mathrm{kg\,m^{-2}}$. For MiRAC-P, the error behaviour is reversed: Small biases and RMSE are found for extremely dry conditions and errors become much larger than the HATPRO-only retrieval for IWV greater than $10\,\mathrm{kg\,m^{-2}}$.

As expected, the synergy performs similarly well or even better than the single instrument retrievals. For IWV below $5\,\mathrm{kg\,m^{-2}}$, the RMSE of the synergy is reduced by 75 % compared to HATPRO while being similar to MiRAC-P. The RMSE of the synergy is also smaller by up to $0.2\,\mathrm{kg\,m^{-2}}$ compared to HATPRO-only when IWV is above $5\,\mathrm{kg\,m^{-2}}$, corresponding to a RMSE reduction of 15–50 %. However, the improvement of RMSE for high IWV is mainly due to the bias reduction from more than $-0.5$ for HATPRO to $-0.1$ to $-0.5\,\mathrm{kg\,m^{-2}}$ for the synergy. When considering the bias-corrected error ($\mathrm{RMSE_{corr}}$),

the synergy shows up to 20 % higher errors than the HATPRO regression retrieval for IWV above $10\,\mathrm{kg\,m^{-2}}$ (not shown). The error reduction compared to MiRAC-P is even higher in this IWV range.

To study the influence of the different retrieval methods and training data sets, we trained one NN with identical settings as used in the final synergy (see Appendix A, Table A1), but included only K–band TBs as input vector. Therefore, the only difference between this NN and the HATPRO regression is the training data (ERA5 vs. Ny-Ålesund radiosondes) and the

retrieval type (regression vs. NN). With this NN, we find that RMSE and biases of the retrieved IWV are similar to those of the HATPRO regression retrieval in almost the entire IWV range (see Appendix B, Fig. B1). Only in very dry conditions (IWV below $2\,\mathrm{kg\,m^{-2}}$, the K–band only NN shows $0.1\,\mathrm{kg\,m^{-2}}$ smaller bias and RMSE. Thus, including the higher frequencies by MiRAC-P dominates the improvement of the error.

## 5.2   Specific humidity profiles

In Fig. 7, the bias and $\mathrm{RMSE_{corr}}$ for the specific humidity profiles of the HATPRO regression retrieval and the synergy NN retrieval are shown with respect to MOSAiC radiosondes. At altitudes below $1500\,\mathrm{m}$ altitude, the $\mathrm{RMSE_{corr}}$ is much smaller for the synergy compared to HATPRO. At the surface, the reduction of $\mathrm{RMSE_{corr}}$ is most prominent, decreasing from $0.5\,\mathrm{g\,kg^{-1}}$ to less than $0.25\,\mathrm{g\,kg^{-1}}$ in absolute terms, and from 30 % to less than 15 % in relative terms (Fig. 7b). Above $1500\,\mathrm{m}$, the $\mathrm{RMSE_{corr}}$ difference between HATPRO and the synergy is marginal and the relative $\mathrm{RMSE_{corr}}$ gradually increases from 25 to

80 % until the top of the retrieval grid ($10000\,\mathrm{m}$). Between the surface and $1000\,\mathrm{m}$, the synergy also shows a much smaller bias ($-0.05$ to $+0.1\,\mathrm{g\,kg^{-1}}$) than HATPRO (0.1 to $0.4\,\mathrm{g\,kg^{-1}}$). The strongest improvement was found near the surface, where the bias is reduced by up to 75 %. Above $1000\,\mathrm{m}$, the bias reduction of the synergy compared to HATPRO is less pronounced: The bias of HATPRO (the synergy) lies between $-0.1$ and $+0.1\,\mathrm{g\,kg^{-1}}$ ($-0.05$ and $+0.15\,\mathrm{g\,kg^{-1}}$). Therefore, combining both instruments is most beneficial in altitudes below $1500\,\mathrm{m}$ in the real-world application.

Because of the different magnitude of specific humidity and the different performances of HATPRO and MiRAC-P over the seasons, we also investigated seasonal differences in error reduction (not shown): In winter (here, 22 October 2019–30 April 2020), the $\mathrm{RMSE_{corr}}$ is lower for both HATPRO and the synergy as also the water vapour amount is lower. However, the relative $\mathrm{RMSE_{corr}}$ of the synergy is similar to the error for the full MOSAiC year in the lowest $1000\,\mathrm{m}$ while the relative error of the HATPRO retrieval is increased. Therefore, the benefit of the synergy in the lower troposphere is even more pronounced.

The synergy also shows smaller errors than HATPRO in the middle and upper troposphere, which was not found for the entire





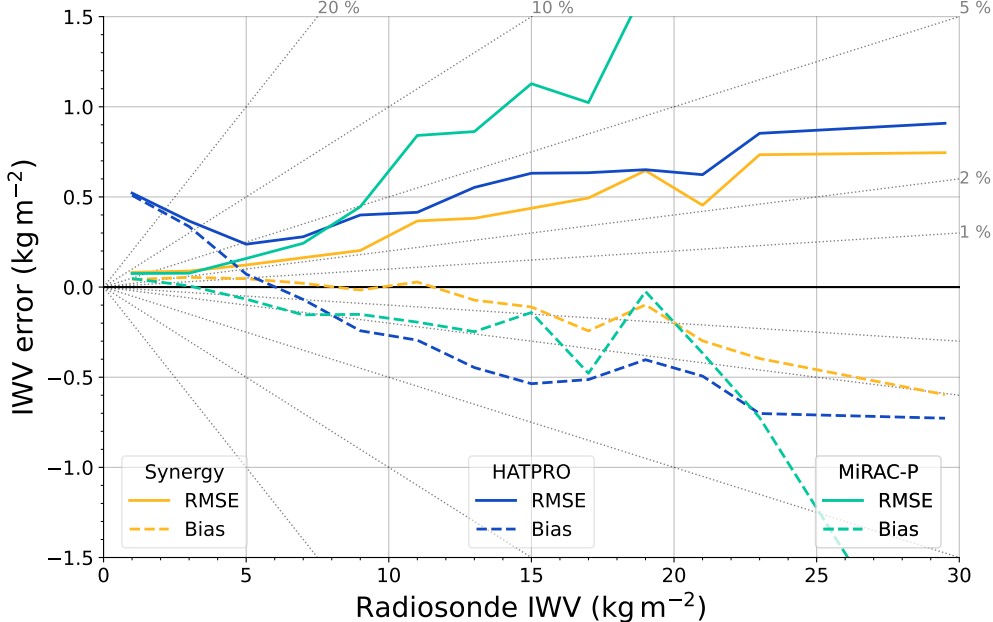

**Figure 6.** RMSE (solid lines) and bias (dashed lines) of IWV retrieved from MOSAiC MWR observations for certain bins of radiosonde IWV (0–2, 2–4, ..., 22–24, 24–35 $\mathrm{kg\,m^{-2}}$). Yellow lines indicate retrieved IWV from the synergy of HATPRO and MiRAC-P, dark blue lines show HATPRO-only and cyan lines show MiRAC-P-only retrievals.

MOSAiC year. The bias reduction of the synergy compared to HATPRO-only is also stronger in winter. In summer (here, 01 May–01 October 2020), the overall picture of the error profiles is similar to the full MOSAiC year, except that the $\mathrm{RMSE_{corr}}$s (relative $\mathrm{RMSE_{corr}}$) for both retrievals are shifted to slightly higher (lower) values. The bias reduction of the synergy compared to HATPRO is also a little less pronounced.

As in Sect. 5.1, to identify whether the error reduction is mainly due to the inclusion of the higher frequencies or due to the different training data and retrieval method, we trained one NN with the same setup as the final synergistic retrieval but used only K–band TBs as input. We applied this K–band-only NN retrieval and the HATPRO regression to the ERA5 and MOSAiC evaluation data sets as in Sect. 4.2 and found that the $\mathrm{RMSE_{corr}}$ was almost identical for both retrievals in all height levels (see Appendix B, Fig. B2b). Only the bias is closer to 0 for the K–band-only NN than for the regression (Fig. B2a). As the results

for both retrieval architectures are mostly similar when using the same input vector (K–band TBs), it follows that the inclusion of the higher frequencies contributes most to the overall error reduction.

     We also investigated the influence of the additional input parameters (2 m temperature, IWV, day of the year) on the retrieved specific humidity profile. In one experiment, we excluded the MiRAC-P TBs from the input vector of the NN but kept the HATPRO TBs, as well as the day of the year, the IWV and the 2 m temperature. The resulting retrieved specific humidity also

shows lower errors than the HATPRO-only regression at the surface (not shown). However, the vertical extent of the benefit is smaller, being mainly confined to the lowest 500 m, compared to the synergistic retrieval including the MiRAC-P TBs. Another



experiment, where we used HATPRO and MiRAC-P TBs, as well as the IWV and day of the year as input but excluded the 2 m temperature, showed higher errors in the lowest 100 m. These experiments demonstrate that the MiRAC-P observations are needed to have a higher vertical extent of the error reduction and that the 2 m temperature effectively reduces errors at the
surface.

To quantify the synergy benefit, it is interesting to analyze not only the error of the retrieved profiles but also their vertical information content. This also offers the opportunity to investigate the impact of the different frequency bands. Thus, we computed the degrees of freedom (DOF) as a measure of the vertical information content for various frequency combinations as described in Sect. 3.3. In Fig. 8, the statistics of the DOF over a 4 % subset of the ERA5 evaluation data set are visualized.
When using only K–band frequencies, the specific humidity profile has about 1.7 DOF. Adding the V–band TBs only has a small effect as these frequencies are hardly sensitive to the water vapour amount. The largest increase in the DOF (from 1.7 to 2.4) is caused by the addition of G–band frequencies to the K–band frequencies. This increase is even more pronounced in cold, dry, and clear sky conditions, where the DOF is increased from 1.9–2.1 to 2.7–3.0 (Fig. 8). In contrast, the DOF hardly improved from 1.6 to 1.8–2.0 in warm and humid conditions. Clear sky scenes are typically associated with cold and
dry conditions during the Arctic winter. The DOF are larger during cold and dry conditions than during warm and humid conditions because the G–band TBs are partly saturated. This means they no longer observe the entire tropospheric column and cannot add as much information. Adding V–band or the 243 and 340 GHz frequencies to K– and G–band TBs only has a minor impact on the DOF distribution.

Ebell et al. (2013) and Löhnert et al. (2009) analyzed the vertical information content of absolute humidity profiles from
ground-based MWRs using K–band TBs at different mid-latitude sites and found 2.4 and 1.6 DOF, respectively. Additionally, Löhnert et al. (2009) obtained 2.7 DOF for a tropical site with a much higher mean IWV. Thus, the DOF depends strongly on the frequencies used to derive the humidity profile and the atmospheric conditions. In the Arctic, humidity profiling is more challenging with K–band frequencies due to the lower sensitivity, which is why the higher frequency observations are needed to obtain similar DOF (see also Fig. 1).

## 5.3 Relative humidity profiles

Relative humidity is an important parameter, particularly for cloud processes, and a desired variable for the modeling community. We computed relative humidity from the retrieved temperature and specific humidity profiles, and radiosonde air pressure. For HATPRO, radiosonde air pressure was not needed to convert the retrieved temperature and absolute humidity profiles to relative humidity. Due to the bias reduction that we achieved with the new NN retrievals in both the retrieved temperature and
specific humidity profiles, we also expect to see lower biases in relative humidity. In the following, we compare the relative humidity bias and RMSE$_{corr}$ of HATPRO and the synergy with respect to the MOSAiC radiosondes, which are shown in Fig. 9.

The bias of the synergy (5 %) is much smaller compared to HATPRO (40 %) in the lowest 1000 m (Fig. 9a). Similarly strong improvements can be found in the lowest 1000 m of the RMSE$_{corr}$ profile (Fig. 9b), where errors are reduced from more than 60 to 15 % at the surface and from 35–45 to 15 % at higher altitudes. Above 2000 m, the RMSE$_{corr}$ of HATPRO and the synergy
are similar (about 20 %), but the bias is closer to 0 % while HATPRO shows a negative bias up to −10 %.



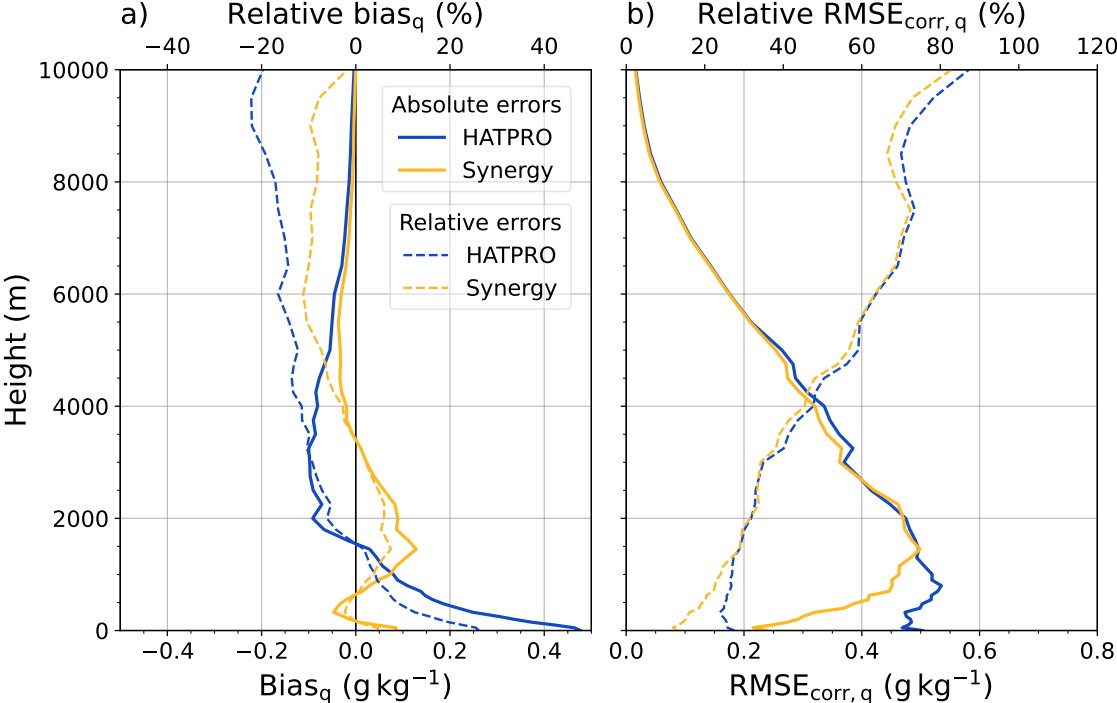

**Figure 7.** Specific humidity $q$ error profiles showing (a) the bias and (b) the bias-corrected RMSE in absolute (solid lines) and relative terms (dashed lines) with respect to MOSAiC radiosondes. Specific humidity errors of the synergy (HATPRO) retrieval are shown in yellow (blue).

In cold and clear sky conditions, where IWV and $2\,\mathrm{m}$ temperatures were below $10\,\mathrm{kg\,m^{-2}}$ and $273.15\,\mathrm{K}$, respectively, and no clouds were detected by Cloudnet as described in Sect. 2.2.2, the bias reduction is even stronger below $1500\,\mathrm{m}$ (Fig. 9a). In warm conditions (IWV $\geq 10\,\mathrm{kg\,m^{-2}}$, $2\,\mathrm{m}$ temperature $\geq 273.15\,\mathrm{K}$), both retrievals perform similarly well, suggesting no benefit of the synergy compared to the HATPRO-only retrieval. If low-level stratus clouds were not respected in the clear
sky detection, the RMSE$_{\mathrm{corr}}$s of the HATPRO retrieval are up to 10 percentage points higher in the lowest $1000\,\mathrm{m}$, while the errors of the synergy only slightly increased (not shown). In general, the relative humidity errors of the synergy are much less sensitive over these two types of atmospheric conditions (or over the seasons, not shown).

## 5.4   Humidity inversion case study

As discussed in the introduction, humidity inversions are important for forming and maintaining clouds. The question arises as
to whether the synergy can resolve humidity inversions better than HATPRO alone. As an example of the humidity profiling capabilities, we present the development of a humidity inversion from 05:05 to 10:49 UTC on 20 November 2019 (Fig. 10). The ERA5 profile shows the upper limit of the NN retrieval profiling capabilities because ERA5 data was used for training. Initially, the HATPRO and the synergy-based specific humidity profiles agree well with the radiosonde observations. In this case, the specific humidity from HATPRO-only is even closer to the radiosonde profile than the one from the synergy below



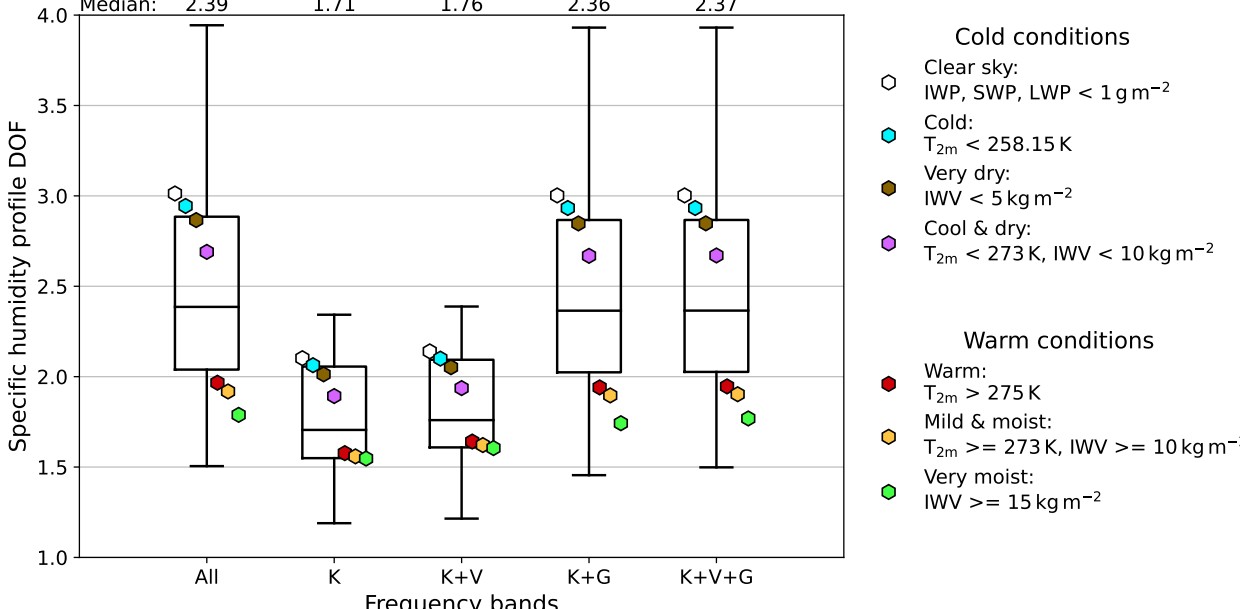

**Figure 8.** Distribution of the degrees of freedom (DOF) over 2803 samples visualized as boxplot for different frequency combinations (all frequencies, K– and V–band, K– and G–band, K–, V– and G–band). The box indicates the interquartile range (IQR, 1–3rd quartile) of the distribution and the horizontal line within the box shows the median. The whiskers extend from below the 1st quartile and above the 3rd quartile by $1.5 \times$IQR, respectively. Additionally, the median DOFs of different atmospheric conditions have been highlighted.

1500 m. At higher altitudes, the synergy agrees better with the radiosonde profile (Fig. 10a). At 10:49 UTC, the strong humidity inversion has formed in the lowest 500 m and is completely missed by the HATPRO retrieval. However, the synergy can capture the approximate location and strength of the inversion well. We also checked other cases and found similar performances of HATPRO and the synergy. However, a more detailed analysis will follow in the future, as this is beyond the scope of this study.

## 6 Conclusions

In this study, we demonstrate the benefit of combining low (22–58 GHz, HATPRO) and high frequency (175–340 GHz, MiRAC-P) microwave radiometer (MWR) observations for humidity profiling and integrated water vapour (IWV) estimates in Arctic conditions. The newly developed Neural Network (NN) retrievals for IWV and for specific humidity and temperature profiles have been applied to synthetic measurements based on ERA5 and real observations from the MOSAiC expedition. Subsequently, they have been evaluated with ERA5 data and MOSAiC radiosondes, respectively, and compared to the retrievals by Walbröl et al. (2022). Retrieved temperature and specific humidity profiles were used to compute relative humidity together with radiosonde air pressure.





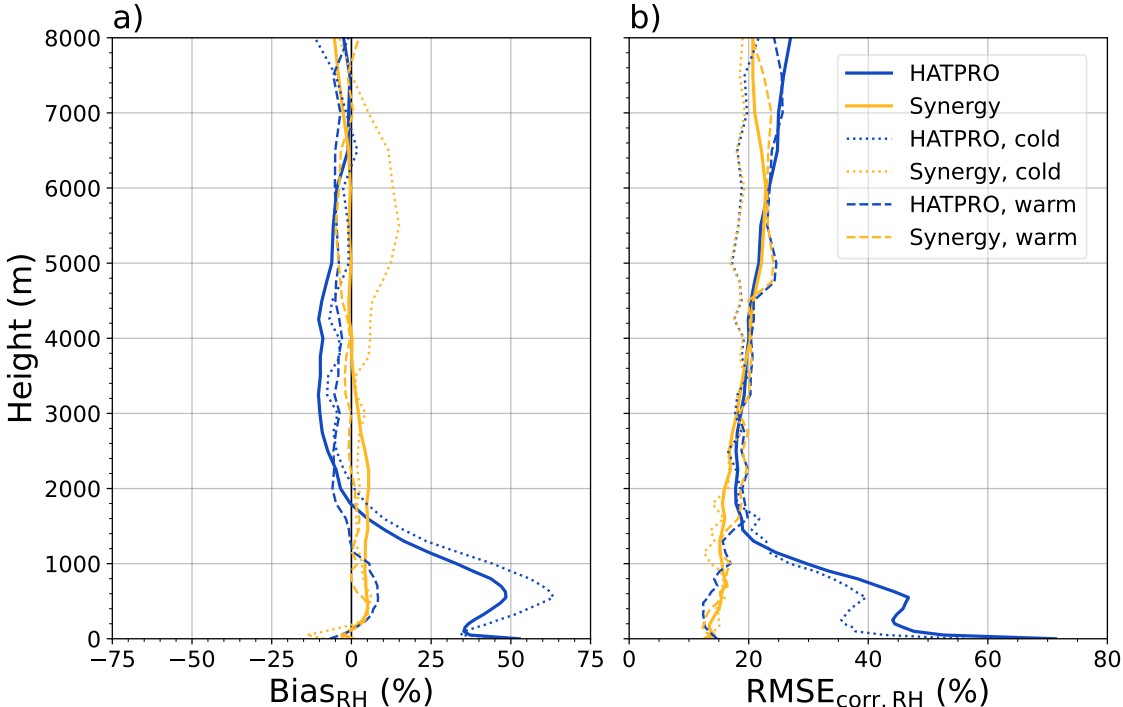

**Figure 9.** Relative humidity error profiles showing (a) the bias and (b) the bias-corrected RMSE with respect to MOSAiC radiosondes. Relative humidity errors of the synergy (HATPRO) retrieval are shown in yellow (blue). Errors are also displayed for different atmospheric conditions: Cold and clear sky (integrated water vapour (IWV) $< 10\,\mathrm{kg\,m^{-2}}$, 2 m temperature (T2m) $< 273.15\,\mathrm{K}$) as dotted lines and warm (IWV $\geq 10\,\mathrm{kg\,m^{-2}}$, T2m $\geq 273.15\,\mathrm{K}$) as dashed lines.

We illustrate the sensitivity of the NN to random perturbations with an ensemble of 20 NNs. The spread of errors over the 20 NNs is generally small, except for specific humidity biases. We selected one NN, whose errors were on the lower end of the spread during the retrieval development, as the final NN. Also in the final evaluation, the final NN denoted one of the smallest

errors of all 20 NNs. In the following paragraphs, we only summarize retrieval errors with respect to MOSAiC radiosondes as these errors are typically larger than the theoretical ones based on the ERA5 evaluation data set: For IWV, the RMSE is about 3–4 % and biases are smaller than 2 % over a wide range of IWV conditions. Specific humidity is overestimated by up to $+0.15\,\mathrm{g\,kg^{-1}}$ at 1500 m relative to radiosondes. In other height levels, the biases are smaller. The bias-corrected RMSE ($\mathrm{RMSE_{corr}}$) is also highest at 1500 m with $0.5\,\mathrm{g\,kg^{-1}}$ (about 30 %). Temperature profile $\mathrm{RMSE_{corr}}$ (biases) from zenith MWR

observations lie between 1.4 and 2.9 K ($-1$ and $+1\,\mathrm{K}$) in the lowest 1500 m. Temperature profiles retrieved from boundary layer MWR observations showed much smaller errors in that height range, which is consistent with the findings of Crewell and Löhnert (2007).

In the next step, we compared the errors of the new synergistic NN retrievals to the single MWR retrievals of Walbröl et al. (2022) to estimate the information benefit. Additionally, we computed the vertical information content of specific humidity



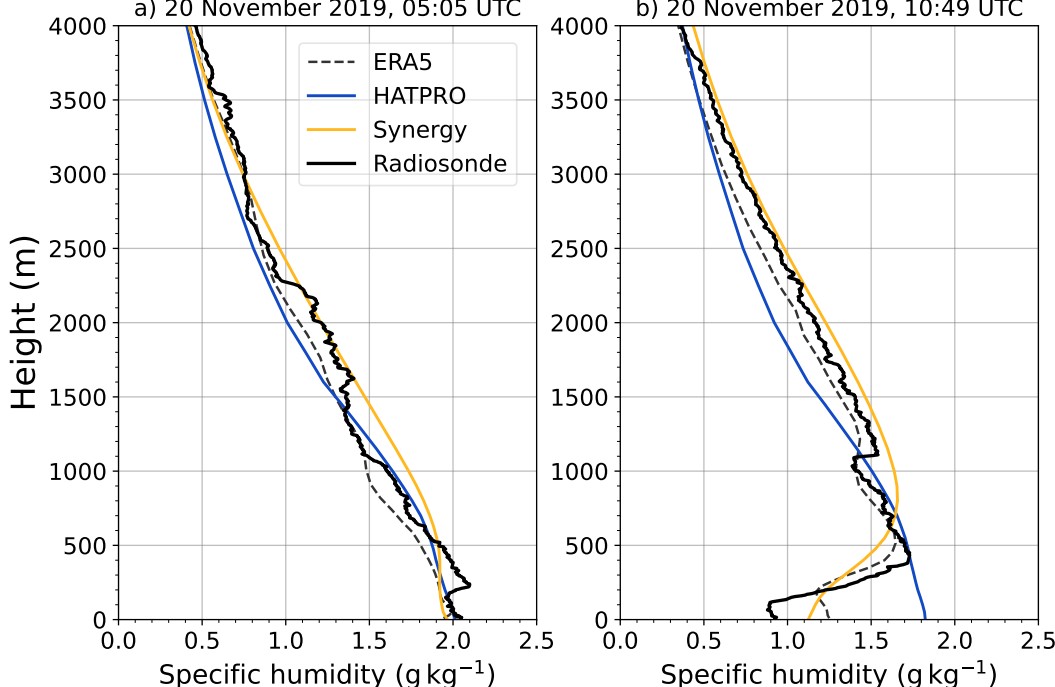

**Figure 10.** Development of a humidity inversion on 20 November 2019 at (a) 05:05 UTC and (b) 10:49 UTC. Specific humidity profiles from radiosondes, ERA5, HATPRO and the synergy are shown as solid black, dashed black, blue, and yellow lines, respectively.

profiles as degrees of freedom (DOF). The information benefit is only shown for MOSAiC observations to obtain the benefit for the real measurements. IWV errors of the synergy are generally smaller than or similar to those of the single MWR retrievals. In cases when IWV is greater than $10\,\mathrm{kg\,m^{-2}}$, the RMSE of the synergy is at least $15\,\%$ smaller than the HATPRO-only retrieval, which is mainly due to the lower biases of the synergy.

For specific humidity profiles, the largest information benefit was found. The combination of HATPRO and MiRAC-P
increased the DOF from 1.7 to 2.4 and reduced the $\mathrm{RMSE_{corr}}$ by up to $50\,\%$. Through the synergy, strong positive biases below $1000\,\mathrm{m}$ could also be reduced by up to $75\,\%$. The benefit is most distinct in the lowest $1500\,\mathrm{m}$ because here, the error reduction is the strongest. In cold and dry conditions, the DOF increase and the error reduction were even more pronounced. We also analyzed the influence of additional NN input parameters ($2\,\mathrm{m}$ temperature, day of the year, and IWV) on the specific humidity profile errors and found that including the $2\,\mathrm{m}$ temperature is important to minimize errors at the surface. Because
of the improvements in specific humidity (and temperature) profiles, the synergy also results in lower relative humidity errors compared to the HATPRO-only retrieval, which is particularly evident in the lowest $1500\,\mathrm{m}$. Additionally, the errors of the relative humidity profiles from the synergy vary much less over different atmospheric conditions than those from the HATPRO-only retrieval.





Current global reanalyses are not able to resolve most small vertical humidity inversions. Therefore, these reanalyses cannot
use the full vertical resolution of radiosondes. The improved humidity profiling of the combined low and high frequency MWRs
has demonstrated the ability to resolve the main inversion well and provide data with a high temporal resolution of 1 second.
Thus, the combination of low and high frequency MWRs seems suitable for studying the representation of humidity profiles in
reanalyses in the Arctic.

Coming back to the research questions listed in Sect. 1, we can conclude:

1. For specific humidity profiles, the bias-corrected RMSE could be reduced by up to 50 %. Bias reductions are partly
        even higher. The information benefit is mainly attributed to the combination of HATPRO and MiRAC-P. The different
        retrieval training data and methods only had a small influence.

     2. The vertical information content in the specific humidity profile was increased by 40 %.

     3. The combination of HATPRO and MiRAC-P frequencies increased the vertical information content in particular during
cold and dry conditions and the least during moist and warm conditions.

HATPROs are used at different sites worldwide (polar, mid-latitude, and subtropical regions). In dry regions (high altitude or
polar sites), the observation network would clearly benefit from an instrument that includes the G–band frequencies for IWV
and humidity profiling (relative and specific humidity) as these frequencies increased the DOF the most. It is planned to install
MiRAC-P at Ny-Ålesund again in 2025 to enhance the continuous atmospheric observations at the German–French research
station AWIPEV. We are confident that adding MiRAC-P to the already installed HATPRO will improve humidity profiling
similarly as demonstrated for the MOSAiC expedition.

In the next step we analyze the ability of the synergy to resolve humidity inversion in more detail. Then, we use the en-
hanced water vapour products from the synergy of HATPRO and MiRAC-P and the radiosonde measurements from MOSAiC
to analyze IWV and specific humidity errors in commonly used satellite products and reanalyses. Further insights into these
situations can be gained from the water vapour mixing ratio profiles derived by the Raman lidar PollyXT that was operated
during MOSAiC on the OCEANET platform (Engelmann et al., 2023). With their high temporal (30 s) and vertical (7.5 m) res-
olution, they are valuable for periods between radiosonde launches in dark and clear sky conditions. We will study the statistics
of humidity inversion characteristics and their differences among the products. We will quantify the errors in downward ther-
mal infrared radiation caused by misrepresentations of the Arctic humidity profile. This can help to identify biases in current
models.

*Code and data availability.* Retrieved profiles of temperature, specific humidity and relative humidity are available on PANGAEA and are
based on brightness temperature observations from HATPRO (Engelmann et al., 2022) and MiRAC-P (Walbröl et al., 2022a). We used the
single instrument retrievals of temperature, absolute humidity and IWV from HATPRO (Ebell et al., 2022) and IWV from MiRAC-P (Walbröl
et al., 2022b) for the benefit estimation. Radiosonde measurements from MOSAiC (Maturilli et al., 2021) and the Polarstern track data (Rex,
2020; Haas, 2020; Kanzow, 2020; Rex, 2021a, b) are also available on PANGAEA. Cloudnet target classification, as well as the low-level



stratus mask and the additional quality flag data, are available and can be accessed via Engelmann et al. (2023), Griesche et al. (2023) and Griesche and Seifert (2023), respectively. Met City observations have been downloaded from Cox et al. (2023). On Zenodo, we published the retrieval training, test and evaluation data (Walbröl and Mech, 2024), the information content estimation output (Walbröl, 2024b), and the ERA5 evaluation data predictions and reference (Walbröl, 2024a). A snapshot of the GitHub repository containing the scripts will be archived at Zenodo. The PAMTRA code can be accessed via Mech et al. (2019b). The simulated brightness temperatures of the two radiosoundings shown in Fig. 1 can be found at Walbröl (2024c).

## Appendix A:  Neural Network retrieval details

As noted in Sect. 3.2, all NNs in this manuscript are multilayer perceptrons (fully connected layers), but some include dropout layers and batch normalization (see Table A1), and have been created with Python's Keras module (contained in Tensorflow, Abadi et al., 2015). The forward propagation of a simple, fully connected NN starts with an input layer whose number of nodes equals the number of components of the input vector. The mathematical operations to propagate to the next layer of the network are similar to multiple linear regression: Each node is multiplied by a randomly initialized weight before being summed up and a bias coefficient is added. Afterwards, the result is used as input to a so-called activation function (e.g., exponential or rectified linear unit, also known as relu). The output of the activation function is then forwarded to each node of the next layer where the process is repeated until the output layer is reached. We always use a linear activation function between the last hidden layer and the output layer. The output layer represents the prediction of the NN and is compared to the truth of the training and validation data sets using a certain loss function (here, mean squared error).

To minimize the loss function, an optimization algorithm (e.g., gradient descent) adapts the weights of each node in a backpropagation process. In this study, we used the Adam optimization algorithm (Kingma and Ba, 2017). The learning rate can be adjusted to reduce or enhance the magnitude of the gradient during backpropagation, leading to slower and smoother or faster and more erratic learning. The NN typically processes a specific number of training data samples, determined by the chosen batch size, before updating the weights. The epoch number determines the maximum number of times the training data set is cycled through. In our retrievals, we activated the EarlyStopping function implemented in Keras that monitors the loss of the validation data set over the epoch numbers. The training was terminated if the validation loss did not improve by more than the minimum delta value for a certain number of epochs (callback patience).

Dropout and batch normalization layers are tools to regularize the NN to make it less prone to overfitting. If batch normalization is set to True for a retrieval (see Table A1), we included a batch normalization layer after each hidden layer. It normalizes the output of the preceding hidden layer so that its mean (standard deviation) is close to 0 (1). The dropout chance noted in Table A1 indicates the chance that the value of a node is set to 0 during training. If the dropout chance is $> 0.0$, we added a dropout layer after each hidden layer or, if applicable, after each batch normalization layer.



**Table A1.** Neural Network settings for each retrieved variable (IWV, specific humidity (q), zenith and boundary layer temperature profiles ($T_{zenith}$, $T_{BL}$)). DOY_1 and DOY_2 are the cosine and sine of the Day Of the Year and T2m is the 2 m air temperature. Details can be found in the text.

| Settings | IWV | q | $T_{zenith}$ | $T_{BL}$ |
|---|---|---|---|---|
| Input vector | TBs at K,G,243,340, DOY_1, DOY_2 | TBs at K,V,G,243,340, T2m, IWV, DOY_1, DOY_2 | TBs at K,V,243,340, DOY_1, DOY_2 | TBs at V, different elevation angles |
| N hidden layers | 2 | 3 | 2 | 2 |
| N nodes per layer | (16,16) | (64,64,64) | (256,256) | (256,256) |
| Activation function | exponential | softmax | relu | linear |
| Dropout | 0.0 | 0.2 | 0.1 | 0.0 |
| Batch normalization | False | True | True | True |
| Batch size | 64 | 256 | 256 | 256 |
| Epoch number | 15 | 100 | 150 | 800 |
| Learning rate | 0.0005 | 0.0005 | 0.0003 | 0.00005 |
| Callback patience | 3 | 30 | 15 | 80 |
| Minimum Delta | 0.001 | 0 | 0 | 0 |

## Appendix B:  Information benefit: Influence of different method

Figure B1 shows the IWV error with respect to MOSAiC radiosondes for the old single instrument retrievals (HATPRO regression, MiRAC-P only NN) and the new NN retrieval. However, in this case, the input vector of the NN consists of K–band TBs only. This demonstrates that the different retrieval method and training data compared to the HATPRO regression is not responsible for the error reduction in dry conditions seen in Fig. 6 and discussed in Sect. 5.1.

Similarly, the specific humidity error profiles for the HATPRO regression and the NN using only K–band TBs are shown in Fig. B2. The RMSE$_{corr}$ of both retrievals is comparable for all height levels but the lower tropospheric bias of the NN, labeled as synergy, is smaller. Therefore, the strong RMSE$_{corr}$ reduction is solely caused by including the higher frequencies in the retrieval. However, the different method and training data set seem to contribute a little to the bias reduction.

*Author contributions.*  AW and MM were involved in the retrieval preparation. AW has performed retrieval development, evaluation, information benefit analysis. All visualizations have been created by AW. AW, KE, HG and SC conceptualized this study and discussed the results. All authors reviewed this manuscript.





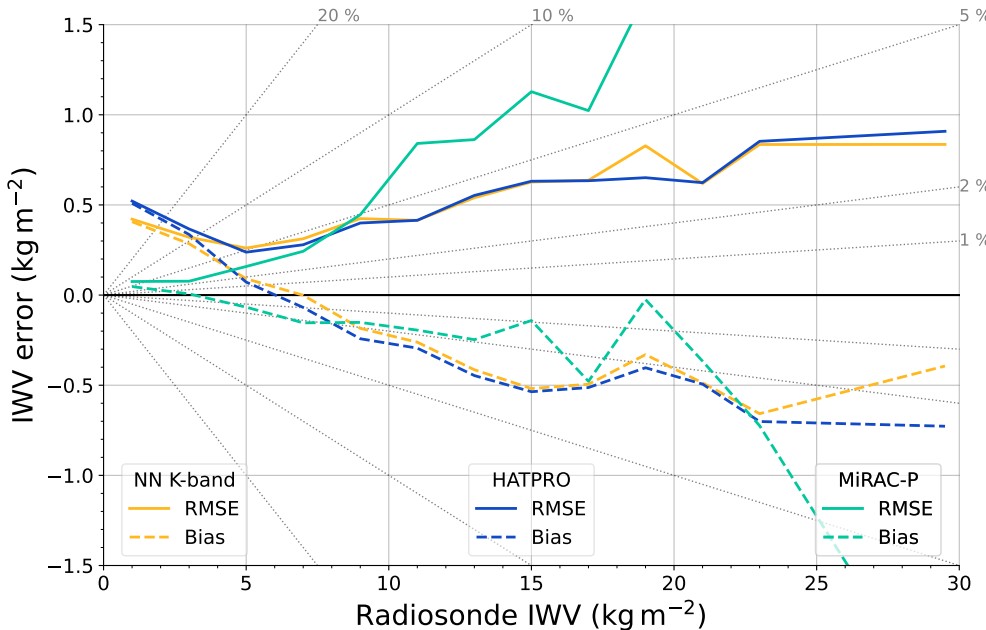

**Figure B1.** As Fig. 6 but using only K–band TBs as input vector to the new NN retrieval.

*Competing interests.* The authors declare that they have no conflict of interest.

*Acknowledgements.* We gratefully acknowledge the funding by the Deutsche Forschungsgemeinschaft (DFG, German Research Foundation)

for the ArctiC amplification: Climate Relevant Atmospheric and SurfaCe Processes, and Feedback Mechanisms (AC)[3] Project Number 268020496 — TRR 172 within the Transregional Collaborative Research Center. Data used in this manuscript was produced as part of the international Multidisciplinary drifting Observatory for the Study of the Arctic Climate (MOSAiC) with the tag MOSAiC20192020 and the Polarstern expedition AWI_PS122_00. We thank all those who contributed to MOSAiC and made this endeavour possible (Nixdorf et al., 2021). The microwave radiometer HATPRO was funded by Federal Ministry of Education and Research (BMBF) under FKZ: 01LKL1603A.

We acknowledge the support from the OCEANET-Atmosphere project, funded by the German Federal Ministry for Education and Research (BMBF) via the SCiAMO project (MOSAIC-FKZ 03F0915A), in which frame the two microwave radiometers were operated. Radiosonde data were obtained through a partnership between the leading Alfred Wegener Institute (AWI), the atmospheric radiation measurement (ARM) user facility, a US Department of Energy facility managed by the Biological and Environmental Research Program, and the German Weather Service (DWD). ERA5 data (Hersbach et al., 2018) were downloaded from the Copernicus Climate Change Service (C3S) Climate

Data Store. The results contain modified Copernicus Climate Change Service information 2022. Neither the European Commission nor ECMWF is responsible for any use that may be made of the Copernicus information or data it contains. This work used resources of the Deutsches Klimarechenzentrum (DKRZ) granted by its Scientific Steering Committee (WLA) under project ID bb1320. Finally, I appreciate the discussions within the working group and with my coauthors.



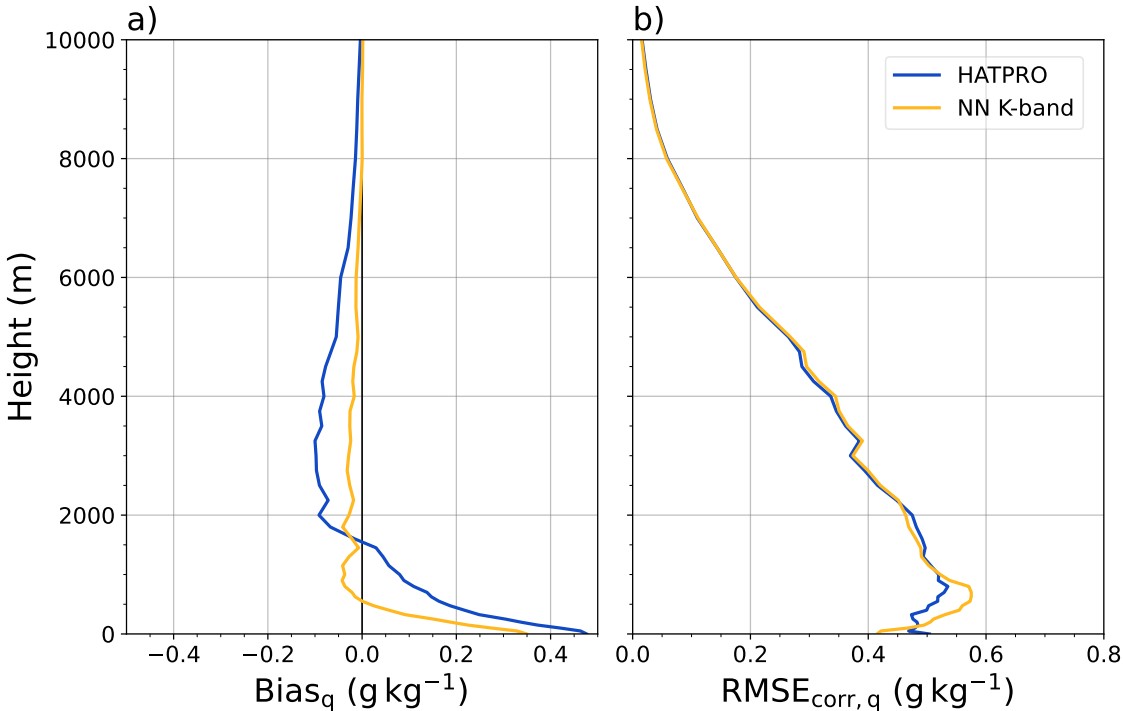

**Figure B2.** As Fig. 7 but showing only the errors for the full MOSAiC year and using only K–band TBs as input vector to the K–band only NN retrieval.

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
