# Peer review of "Combining low and high frequency microwave radiometer measurements from the MOSAiC expedition for enhanced water vapour products"

_EGUsphere, 2024_

## Author Comment (AC1)

**University of Cologne**

Cologne, 25 July 2024

**Author's response:**

Dear AMT editorial team,

we have revised the manuscript "Combining low and high frequency microwave radiometer measurements from the MOSAiC expedition for enhanced water vapour products" according to the comments of the reviewers (see detailed response below). The revised version includes additional analyses on the resolution of the single-instrument and synergistic retrievals. We improved the motivation and explanation of the retrieval setup and updated the relative humidity computation.

Neither this manuscript or substantial parts of it have been published elsewhere in English or any other language, nor is it presently under consideration for publication by any other journal.

Sincerely,

Andreas Walbröl

and Meteorology MSc. Andreas Walbröl Phone: +49 (0)221 470-3678 E-mail: a.walbroel@uni-koeln.de

**Institute for Geophysics**

Pohligstr. 3 50969 Cologne Germany

**Reply to reviewer 1:**

We thank the reviewer for the supportive review of the manuscript. Below, we repeat the reviewer's comments in black and write our responses in blue. The line numbers in the lineby-line responses are valid for the revised manuscript.

The work presents retrievals of vertically resolved temperature, humidity, and integrated water vapor (IWV) from combined measurements of 2 microwave radiometers, including frequencies at or higher than 183 GHz. The retrievals were trained with ERA5 data from the European Centre for Medium-Range Weather Forecasts (ECMWF) and synthetic MWR observations simulated with a radiative transfer code. Retrievals were evaluated against ERA5 and radiosondes launched during MOSAiC. The combined retrievals noticeably improved the IWV and the profiles compared to retrievals where only lower frequencies were used.

**General comment**

The paper is generally well written and complete in its explanation, my main question is of a general nature and goes back to the broader issue of the advantage of using radiometers vs. reanalysis for profiling. As the authors state in the introduction current reanalysis have difficulties in accurately representing the Arctic winter atmosphere but, in this context the paper does not shed any light on whether this combination of frequencies provides an improvement over the reanalysis. For example, in figure 5, rather than seeing errors related to MWR-MOSAIC and MWR-ERA5, it would be better to show MWR-MOSAIC and ERA5-MOSAIC. This way we can see if the retrievals offer a better representation of the low troposphere in the Arctic compared to ERA5.

We understand that it is an important point to assess whether the radiometers perform better than the reanalysis. However, during the MOSAiC expedition, the radiosondes have been transmitted to the Global Telecommunication System and have been assimilated by ERA5. Thus, the performance of ERA5 during the time of the MOSAiC expedition is strongly influenced by the radiosondes launched from RV Polarstern. A comparison of the MWR retrievals with ERA5 for this period would not be representative of ERA5's "normal performance", e.g., if no radiosonde had been assimilated. For clarification, Fig. 5 illustrates the best possible accuracy by using ERA5 data from the years 2001, 2006, 2011 and 2015, which were left out when the retrieval algorithm was trained. An evaluation of the performance of reanalysis is difficult as the quality of the reanalysis is improved due to the additional MOSAiC radiosondes, which were not available during other times. We now highlight this point in our conclusion: "As reanalyses assimilated the MOSAiC radiosonde observations, this comparison likely does not reflect the true performance of the reanalyses in the central Arctic." (line 522-523)

Similarly, looking at Fig. 10 b, it appears that ERA5 still does a better job than the synergy to represent the humidity vertical structure. Therefore, my question is, why not just use ERA5? It appears that even after all the improvements shown in this work, radiometric profiles are still not good enough to provide useful vertically resolved humidity. Are they just a smoother version of the reanalysis dataset used to train them? Or they actually improve on the reanalysis biases?

Also here, the effect of the assimilation of the radiosonde data into ERA5 can be seen in the ERA5 profile. It is not the aim to improve the reanalysis bias in the presence of radiosonde data. The goal of this study was to explore the IWV and humidity profiling benefits when combining the low- and high-frequency observations. Based on the suggestion of reviewer 2, we added a new figure (Fig. 9), which clearly illustrates the better resolution of the synergy. An application example could be that the synergy can provide useful insight into the Arctic humidity profile at sites where no radiosonde are launched on a regular basis. We added this example to the conclusions: "The low specific humidity profile errors give us confidence that the synergy is suitable for gaining insights into the general structure of Arctic humidity profiles (i.e., inversions). However, a detailed analysis of the ability of the synergy to identify humidity inversions is still missing. [...] With the considerable specific humidity profile improvements of the synergy compared to HATPRO, the question arises how well humidity inversions, which are important for cloud formation and maintenance, are captured. This question will be answered with a statistical analysis for the entire MOSAiC period." (line 517-519, 523-525)

**Reply to reviewer 2:**

We thank the reviewer for the supportive review of the manuscript. The comments addressed some important points that needed to be added to our study. Below, we repeat the reviewer's comments in black and write our responses in blue. The line numbers in the line-by-line responses are valid for the revised manuscript.

This is a well-written paper that is easy to follow and clearly articulates its goals and its approach to investigating them. Overall, this paper is appropriate and suitable for publication in AMT after some relatively simple issues have been addressed. Unfortunately, I feel that this includes some small data analysis which raises the threshold from minor to major revisions. Still, these revisions should not be difficult to implement.

**> Thank you for supporting this study.**

Line 191: The authors state that they use NNs "because they can deal better with the nonlinear relationship between IWV and TB measurements in the G-band." I feel that this statement needs more supporting justification. What is this comparison being made to (i.e. NNs are better than what other technique?) Why and how are they better? One of the chief advantages of NNs is that, after they have been trained, they are computationally very quick to perform retrievals as compared to physical/OE methods. However, physical methods have a significant advantage in that they are better suited to retrieving observations of conditions that are not represented in the training dataset, which could be important in a changing arctic. I am not suggesting that the authors redo their entire analysis with a brand new retrieval method; however, I feel that they need to more strongly justify why they chose this method.

Thank you for pointing out that this needs to be better explained in the manuscript. We added that the comparison was made with respect to regression: "...we use Neural Networks (NNs) because they can deal better with the nonlinear relationship between

IWV and TB measurements in the G-band compared to regression. During the development of the MiRAC-P-only retrieval (Walbröl et al., 2022), tests showed that the IWV retrieved with a multiple nonlinear regression had a significantly higher spread than when retrieved with NNs." (line 196-200)

We also improved our argumentation why we preferred NN over a physical approach by adding:

"However, at the high frequencies of MiRAC-P, the scattering of radiation by frozen hydrometeors cannot be neglected and may therefore introduce uncertainties in the radiative transfer calculations needed for the forward simulation F(x). The retrieval would require assumptions on hydrometeor properties (concentration, size, shape, orientation) or further hydrometeor observations, making it dependent on the availability of such observations." (line 188-192)

Line 210: Why was this grid chosen? There is growing awareness that the selection of a retrieval grid has a notable influence on DOF and vertical resolution (see Loveless et al. 2023 doi:10.1109/IGARSS52108.2023.10283250). Did you assess any of the influences that the grid choice may have had on the results?

During the development of the retrieval, we also experimented with different grids and noticed slight changes. We selected this height grid to keep it consistent with that from the single-radiometer retrievals, which is also used in the standardized HATPRO retrievals [1,2] as also highlighted in the text:

"For the retrieval development and evaluation, atmospheric profiles have been interpolated onto the same height grid used in the standard HATPRO retrieval (Löhnert, 2023; Marke et al., 2024) and in Walbröl et al. (2022), ranging from 0 to 10000 m with the vertical spacing increasing from 50 m at the surface to 500 m at the top."

[1]: Löhnert, U. (2023): Ground-based microwave radiometer reprocessing mwr\_pro (Version v04). Zenodo. https://doi.org/10.5281/zenodo.7973553.
[2]: Marke, T. et al. (2024): A Python package for processing microwave radiometer data. Journal of Open Source Software, 9(98), 6733, https://doi.org/10.21105/joss.06733.

Lines 265 and onward: I am approaching this problem from the perspective of an observationalist, not a modeler or a strong user of ERA5. With that in mind, it seems intuitive that an NN that is trained on ERA5 will produce profiles that show stronger agreement with ERA5 than with an external data source not just because of the uncertainties of the radiosondes or the spatiotemporal matching issues (though those concerns are important) but because the NN is implicitly including all of the known biases and uncertainties of ERA5. To me, the comparison to radiosondes is more interesting and important than the comparison to ERA5, but it is that reanalysis comparison that gets the stronger focus of attention. One step forward may be stronger justification of why the ERA5 comparisons are important.

We agree that the retrieved profiles rather resemble an ERA5 profile than a measured radiosounding. By including also some radiosondes (about 5 % of all MOSAiC radiosondes) in the retrieval validation procedure, we reduced the risk of including biases from ERA5 in our retrieval.

The comparison to the ERA5 evaluation data (years 2001, 2006, 2011, 2015) serves as an

estimate of the best performance. Of course, for real-world applications, the comparison with the radiosondes gives a much better estimate. This is also written in the manuscript: "The retrieval evaluation with respect to the ERA5 data allows us to assess the retrievals' theoretical best performance... " (line 274-275)

Line 305: It sounds like the radiosonde analysis was performed with respect to sondes that had been interpolated to the retrieval grid, and not smoothed to account for the influence of weighting functions. Given that you have averaging kernals for a random subset of the cases, it would be possible to vertically smooth the sondes. This would permit an analysis of the bias and RMSE relative to sondes that would remove the influence of the large vertical gradients and provide a more direct comparison between the statistics calculated relative to sondes versus those from ERA5.

This certainly is an important point to mention in the manuscript, but we think that the actual errors to the true atmospheric state, which is assumed to be provided by the radiosonde, are more useful to the reader. We added a comparison of the smoothed radiosonde specific humidity profiles to the appendix (Appendix C).

Lines 411 and onward: the DOF is an important quantity for evaluating the information content of a retrieved profile. Also important is the vertical resolution, which is easily obtained by scaling the vertical grid spacing by the inverse of the averaging kernal. As you already have these pieces of information, a small discussion of the changes in the vertical resolution brought on by the synergy would be achievable and relevant to this paper's aims.

Figure 9. Vertical resolution of the specific humidity profiles estimated with the mean Averaging Kernel over the 2803 samples and vertical grid spacing for all frequencies (yellow) and for K–band only (blue).

Line 432: Why did you opt for radiosonde pressure instead of a hypsometrically-derived pressure profile from the T and specific humidity profiles? This would allow for a self-contained measurement of RH without relying on the relatively-infrequent sonde observations.

Good point. We changed the relative humidity computation. We updated the relative humidity data set and the corresponding figure (former Figure 9, now Figure 10).

Line 448: The paper may be stronger if this particular section is removed. As it stands, it introduces a somewhat counterintuitive example (the synergy makes things worse) then promises to investigate further in a future paper. It may be best if the entire discussion is held to that future paper so that it can be investigated in detail and the nuances can be explained.

We may not have chosen a good example (it was somewhat arbitrarily selected) in this case to demonstrate the capability of the synergy. Thus, we agree to remove this section and changed the outlook for our upcoming study.

Line 486: Here is a good example of why the inclusion of a vertical resolution discussion is important. Improvements are seen in the lowest 1500 m, but I'm guessing that even with the synergy and the non-zenith stares, the true vertical resolution is on the order of 1500 m or worsegiven the broad weighting functions of the microwave band. Noting the changes in the vertical resolution would help better attribute the causes of these improvements.

This is indeed a good example that demonstrates the importance of adding the vertical resolution estimate. We added the following to the conclusions: "At these heights, the synergy enhanced the effective vertical resolution of the specific humidity profile by a factor of up to 2 compared to the HATPRO-only retrieval (from 1200 m to 600 m)." (line 496-497)

---

## Author Response (AR2)

**University of Cologne**

[Figure]

Cologne, 11 September 2024

**Author's response:**

**Institute for Geophysics**

**and Meteorology**

**MSc. Andreas Walbröl**

Phone: +49 (0)221 470-3678

E-mail: a.walbroel@uni-koeln.de

Pohligstr. 3

50969 Cologne

Germany

Dear AMT editorial team,

thank you for accepting our manuscript "Combining low and high frequency microwave radiometer measurements from the MOSAiC expedition for enhanced water vapour products" for publication. As final changes, we have updated the reference of the codes for the reproduction of the figures.

On behalf of all authors and with best regards,

Andreas Walbröl